# Genome-wide deposition of 6-methyladenine in human DNA reduces the viability of HEK293 cells and directly influences gene expression

Julian Broche[1,3], Anja R. Köhler[1], Fiona Kühnel[1], Bernd Osteresch[2], Thyagarajan T. Chandrasekaran[1], Sabrina Adam[1], Jens Brockmeyer[2] & Albert Jeltsch [1][✉]

While cytosine-C5 methylation of DNA is an essential regulatory system in higher eukaryotes, the presence and relevance of 6-methyladenine (m6dA) in human cells is controversial. To study the role of m6dA in human DNA, we introduced it in human cells at a genome-wide scale at GANTC and GATC sites by expression of bacterial DNA methyltransferases and observed concomitant reductions in cell viability, in particular after global GANTC methylation. We identified several genes that are directly regulated by m6dA in a GANTC context. Upregulated genes showed m6dA-dependent reduction of H3K27me3 suggesting that the PRC2 complex is inhibited by m6dA. Genes downregulated by m6dA showed enrichment of JUN family transcription factor binding sites. JUN binds m6dA containing DNA with reduced affinity suggesting that m6dA can reduce the recruitment of JUN transcription factors to target genes. Our study documents that global introduction of m6dA in human DNA has physiological effects. Furthermore, we identified a set of target genes which are directly regulated by m6dA in human cells, and we defined two molecular pathways with opposing effects by which artificially introduced m6dA in GANTC motifs can directly control gene expression and phenotypes of human cells.

[1] Institute of Biochemistry and Technical Biochemistry, Department of Biochemistry, University of Stuttgart, Allmandring 31, 70569 Stuttgart, Germany. [2] Institute of Biochemistry and Technical Biochemistry, Department of Food Chemistry, University of Stuttgart, Allmandring 5b, 70569 Stuttgart, Germany. [3] Present address: Institute of Medical Genetics and Applied Genomics, University of Tübingen, Calwerstr. 7, 72076 Tübingen, Germany. [✉]email: albert.jeltsch@ibtb.uni-stuttgart.de

DNA, the molecule of inheritance in biology, can be chemically modified to establish so-called epigenetic signals[1–3]. This includes the methylation of DNA at the C5- and N4-atoms of cytosine and the N6 of adenine. It has been established since decades, that DNA methylation plays numerous important roles in bacteria, and cytosine-C5 methylation is essential for mammalian development[1–3]. Strikingly, the role and existence of N6-methyl-2′-deoxyadenosine (m6dA) in human DNA have remained very controversial. Originally thought to be absent, this modification has been rediscovered in 2015 at low levels in several higher eukaryotes[4–8]. This finding was followed by several publications reporting its presence in the DNA of human cells and other mammals and documenting its impact on gene regulation and diseases (reviewed in[9–11]). Moreover, comparative studies revealed conserved m6dA patterns in different species[12]. Highest m6dA levels of up to 1000 ppm in human cells were observed in heterochromatic regions[13], but m6dA levels were observed to change strongly during development and differ between tissues and genetic elements[14,15]. M6dA was found to be depleted from gene exons[8], while in other studies it was reported to be enriched in gene bodies and positively associated with gene expression in human cell lines[13,16]. The presence of m6dA in gene promoters was reported to stimulate expression of BDNF in neurons and mediate fear conditioning in mice responding to early life stress[17,18]. However, m6dA was also reported to prevent chromatin binding of the SATB1 factor[13] and to correlate negatively with LINE retrotransposon activity in embryonic stem cells and in the brain of adult C57/BL6 mice following exposure to chronic stress[16,19]. Moreover, altered m6dA patterns were observed in cancers[13,20,21]. Other studies reported m6dA in mitochondrial DNA where it was connected with mitochondrial stress and reported to function as a transgenerational epigenetic signal[22,23]. Finally, m6dA has been implicated in DNA repair[24].

On the other hand, the presence and biological relevance of m6dA in human DNA is discussed very controversially. Different research groups have provided evidence arguing against the presence of large amounts of this modification in human DNA, highlighting several technical problems and potential artifacts[11,25–30]. Moreover, the functional relevance of the low levels of m6dA found by some groups in human DNA has been questioned[11,31,32]. It was found that m6dA could be introduced into the DNA accidentally through DNA replication by incorporation of m6dATP, generated via recycling of m6A-containing RNA or m6dA containing DNA from symbiotic bacteria[11,28–30,32]. Moreover, m6dA could be introduced into the DNA as side activity of RNA-modifying enzymes[11,23,33,34]. In fact, despite the investigation of several candidate enzymes like N6AMT1, METTL3-METTL14, and METTL4, so far, none of them has been accepted by the field as human DNA-(adenine N6)-methyltransferase (N6-MTase). The different modes of m6dA incorporation into the human genome and the lack of a bona fide human N6-MTase shed questions on the biological role of m6dA in human DNA in general. Hence, additional work applying novel research approaches is urgently needed in the field to clarify the potential biological roles of m6dA in human cells.

We have devised a stepwise approach to investigate the potential biological roles of m6dA in human genomic DNA and then identify and validate potential regulatory targets and mechanisms. Our starting consideration was that it is very difficult to detect endogenous m6dA and investigate its biological roles directly, as illustrated by several previous studies with partially contradictory results mentioned above. Therefore, instead of focusing on the detection of endogenous m6dA and direct investigation of its potential biological roles, we used an orthogonal approach and expressed highly active bacterial N6-MTases in human cells to introduce high levels of m6dA within the sequence motifs specific for the corresponding N6-MTases. This allows to study the effects of the global introduction of m6dA on cell viability, proliferation and gene expression without the difficulties related with low m6dA levels and unclear mechanisms of their enzymatic introduction. Our data reveal mild but clearly detectable reductions in cell viability by the genome-wide introduction of m6dA, in particular after methylation in GANTC motifs. By gene expression analysis, we identified 99 genes that are regulated by m6dA in a GANTC context. Genes upregulated by m6dA showed reduction of H3K27me3 suggesting that PRC2-dependent H3K27me3 deposition can be reduced by m6dA in GANTC motifs. Genes downregulated by m6dA showed an enrichment of JUN family transcription factor (TF) binding sites, which contain 5′-T and 3′-A flanked GANTC sequences in their recognition motif (TGANTCA), suggesting that adenine methylation of the GANTC motifs within these sites can reduce the recruitment of JUN TFs to target genes. Thus, our study documents that global, genome-wide introduction of m6dA in human DNA has biological effects. Furthermore, we defined two molecular pathways by which m6dA can directly control phenotypes of human cells and identified target genes which are regulated by DNA adenine methylation. These results set a common ground for future research on the role of m6dA in human DNA.

## Results

In this work, we aimed to explore the effects of global genomic N6-deoxyadenine DNA methylation on the viability and proliferation of human cells by expressing different N6-MTases in HEK293 cells. Over the past years, several enzymes have been proposed to be responsible for adenine methylation in human DNA, but additional studies will be required to further validate their functionality in vitro and in vivo[11]. For this reason, we decided to employ for our study two well-characterized bacterial N6-MTases, EcoDam, and CcrM, which both exhibit high catalytic activity and specificity for DNA methylation at GATC and GANTC sites, respectively[1,3]. As control, catalytically inactive EcoDam D181A and CcrM D39A mutants were generated containing an aspartate-to-alanine exchange in a catalytically essential DPPY motif of N6-MTases[35]. We generated a set of stable HEK293 cell lines expressing CcrM or EcoDam, in each case either the wildtype enzyme (WT) or mutant, under the control of a doxycycline (dox)-inducible TRE3G promoter (Fig. 1a). By this means, the N6-MTases could be expressed for several weeks, giving them time to introduce N6-adenine methylation into the genomic DNA. The co-expression of a fluorophore over an internal ribosomal entry site (IRES) allowed us to identify and enrich the cells actively expressing the N6-MTase in flow cytometry and FACS experiments.

**EcoDam and CcrM are highly active in human HEK293 cells.** As first step, we wanted to verify the activity of the N6-MTases in vivo by inducing them with dox for up to 10 days. N6-MTase expressing cells were enriched by FACS of GFP-positive cells. After 3 and 10 days of dox treatment, the genomic DNA (gDNA) of treated and untreated cells was extracted and subsequently digested with different m6dA-sensitive restriction enzymes, viz. HinfI cleaving unmethylated GANTC sites, DpnII cleaving unmethylated GATC sites and DpnI cleaving methylated GATC sites (Suppl. Fig. 1a). The gDNA was efficiently cleaved by HinfI in uninduced state of CcrM ('no dox') revealing a lack of m6dA at most of the genomic GANTC sites (Suppl. Fig. 1b). However, upon induction of WT CcrM, HinfI cleavage was blocked almost entirely, demonstrating the high methylation efficiency of the enzyme which introduced an almost complete genome-wide methylation of GANTC motifs. In contrast, DpnI showed no

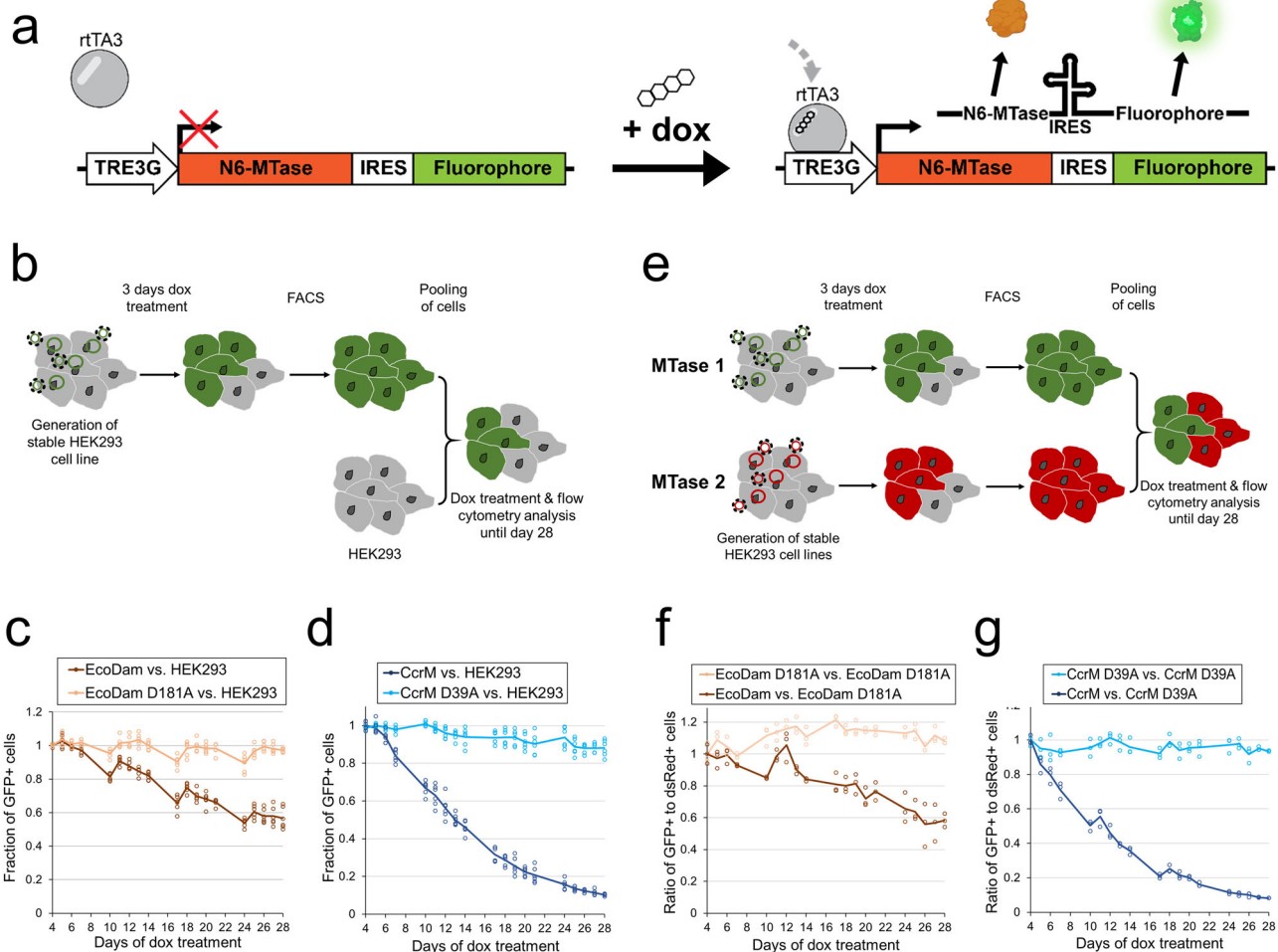

**Fig. 1 Cell viability effects of the genome-wide deposition of m6dA by expression of bacterial N6-MTases. a** Design of the N6-MTase expression constructs which were stably integrated into the genome of HEK293 cells. The reverse tetracycline-controlled transactivator 3 (rtTA3) binds the TRE3G promoter in the presence of dox, thus activating the expression of the N6-MTase. A fluorophore is co-expressed over an internal ribosomal entry site (IRES). **b** Schematic depiction of the experimental workflow used for the competitive proliferation assay of N6-MTase versus HEK293 cells. **c, d** Flow cytometry results of the competitive proliferation assay for EcoDam (WT and D181A mutant), or CcrM (WT and D39A mutant), against HEK293 cells. The fraction of GFP-positive (GFP$^+$) cells relative to non-fluorescent cells was calculated and normalized to day 4 of dox treatment. **e** Schematic depiction of the experimental workflow used for the competitive proliferation assay of N6-MTase 1 (GFP$^+$) versus N6-MTase 2 (dsRed$^+$). **f, g** Flow cytometry results of the competitive proliferation assay for EcoDam or CcrM variants (GFP$^+$) versus the corresponding inactive mutants (dsRed$^+$). The fraction of GFP$^+$ cells was calculated and normalized to day 4 of dox treatment.

activity on the gDNA with or without CcrM induction and DpnII was fully active, indicating the absence of methylation at GATC motifs after CcrM expression in cells. In comparison, the induction of the catalytically inactive D39A mutant did not lead to any detectable changes in the digestion pattern indicating a lack of m6dA occurrence at GANTC sites in the genome (Suppl. Fig. 1d). Correspondingly, after the induction of WT EcoDam, DpnII cleavage was efficiently blocked genome-wide, and vice versa most of the gDNA was digested by DpnI after 3 or 10 days of dox induction (Suppl. Fig. 1c). Both these findings indicate the global genome-wide methylation of GATC motifs. Lack of protection of this gDNA against HinfI cleavage indicates absence of methylation at GANTC motifs after EcoDam expression in cells. Again, no detectable change in the methylation of GATC sites was observed after expression of the catalytically inactive EcoDam D181A mutant (Suppl. Fig. 1d). Interestingly, for both N6-MTases, almost complete global methylation of the target motifs was already achieved after 3 days of dox induction which remained stable up to day 10 in the presence of dox. With these

results, the strong and global activity of the WT N6-MTases within human cells was confirmed, while the inactive mutants did not show detectable activity, thus serving as appropriate negative controls.

**Genomic introduction of m6dA reduces the proliferation of HEK293 cells.** Next, we aimed to investigate the potential effects of genome-wide adenine methylation on cell proliferation. Therefore, CcrM and EcoDam, as well as their respective inactive mutants, were induced for 3 days with dox, followed by FACS enrichment of GFP-positive (GFP$^+$) cells (Fig. 1b). The sorted cells were then pooled in a 1:1 ratio with untransduced HEK293 cells, not expressing any fluorophore (GFP$^-$). The cells were then re-cultivated in a cell culture vessel and propagated in the presence of dox. This way, the competitive proliferation of the stable cell lines and uninfected HEK293 cells could be directly compared. Every 1–3 days, the ratio of GFP$^+$ to GFP$^-$ cells was measured using flow cytometry. All experiments were conducted

in two biologically independent replicates each including three repeats treated in parallel. Neither EcoDam D181A, nor CcrM D39A did induce a noteworthy shift in the relative amount of GFP⁺ cells after expression for up to four weeks, indicating that the continuous long-term expression of these proteins and dox treatment did not compromise the viability of HEK293 cells under our conditions (Fig. 1c, d). In contrast, the presence of active EcoDam led to a gradual decline in GFP⁺ cells, with an overall reduction of 43% GFP⁺ cells after 28 days of dox treatment (Fig. 1c). This observation pointed to an effect of the m6dA modification on cell proliferation in the GATC sequence context. Strikingly, the expression of CcrM resulted in an even more accelerated reduction of GFP⁺ cells. The relative amounts of WT CcrM-expressing cells were already halved after ~13 days of dox treatment, and reduced by 90% after 28 days (Fig. 1d).

In order to further rule out any unanticipated side-effects of the dox treatment on the infected and uninfected cells, the competitive proliferation assay was repeated with an adjusted setting. This time, the HEK293 cells expressing WT EcoDam or CcrM were combined with cells expressing the corresponding inactive mutant and co-expressing dsRed instead of GFP over an IRES. Thus, the relative change of GFP⁺ to dsRed⁺ cells could be tracked after pooling the cells in a 1:1 ratio (Fig. 1e). As controls, inactive mutant populations co-expressing GFP or dsRed in a 1:1 ratio were used (Fig. 1f, g). In both cases, the relative ratios of GFP⁺ and dsRed⁺ cells did not change in the mutant controls, indicating that the expression of the different fluorophores did not affect the results. However, cells expressing WT EcoDam were depleted to a similar extent (−43%) as previously. Very similar dynamics were also observed for CcrM, where the WT CcrM-positive cells (GFP⁺) were reduced by 92% after 28 days. Based on these results we conclude that the reduced proliferation of the cells expressing the wild-type N6-MTases is directly related to their catalytic activity, hence it reflects the deposition of m6dA in the genome.

**Combined expression of EcoDam and CcrM in HEK293 cells leads to increased m6dA levels.** As the expression of individual N6-MTases resulted in a decrease in cell proliferation, we next aimed to further increase the genomic m6dA levels. This was achieved by generating stable cell lines, which express both N6-MTases simultaneously in the presence of dox. To this end, the already established EcoDam WT and mutant cell lines (GFP⁺) were re-transduced with constructs containing either WT or mutant CcrM, both co-expressing dsRed over an IRES (Fig. 2a). The different combinations of the N6-MTases should enable to introduce varying levels of m6dA into the genomic DNA, with WT CcrM combined with WT EcoDam expected to introduce the highest methylation levels, because GANTC and GATC motifs are targeted. The methylation activity of both induced N6-MTases was again validated by restriction digestion of gDNA using different enzymes. The results of this experiment illustrate that the co-expression of WT EcoDam and WT CcrM led to the characteristic digestion pattern of the gDNA that confirms the global methylation of GATC and GANTC sites (Suppl. Fig. 2a, b).

To confirm the results of the gDNA restriction digestion experiments, we quantified the m6dA levels in the gDNA extracted from the generated cell lines by HPLC-MS/MS using external dA- and m6dA standards (Suppl. Fig. 3). As shown in Fig. 2b, the m6dA levels in untransduced HEK293 cells were at the detection limit, thus supporting the general observation that adenine methylation does not occur extensively in most human cell types including HEK293 cells. Many additional technical and biological controls would be needed to assess if this represents true biological DNA methylation but solving this question was

not the aim of our work. Similarly, almost no adenine methylation was detected in cells expressing the inactive mutants of CcrM or EcoDam. Intriguingly, after 3 days of dox induction of the WT EcoDam or WT CcrM, a strong increase in m6dA levels was observed up to 12,000 and 14,600 ppm m6dA vs. dA, respectively (corresponding to 1.20% and 1.46% of m6dA, respectively). Given that the human genome has a GC-content of 40.87% (Hg38)[36], 1.235% of all adenines would statistically occur in each of the GATC or GANTC sequence contexts. Hence, the measured m6dA levels document a complete genome-wide methylation of all target sequences which is in line with the gDNA restriction digestion results described above. Longer induction of EcoDam or CcrM for up to 10 days did not result in a noticeable change in the m6dA concentration detected in HPLC/MS when compared to the results obtained after 3 days of dox treatment (Fig. 2b). This observation also agrees with the results of the gDNA digestion studies described above, indicating that the maximum of adenine methylation has already been reached after 3 days of dox. The HPLC-MS data also revealed that the co-expression of WT CcrM and WT EcoDam resulted in approximatively additive methylation levels when compared with the cells expressing only one N6-MTase, indicating that both N6-MTases acted independently of each other at most target sites.

Next, the effects of the higher genomic m6dA levels introduced by the combined expression of WT EcoDam and WT CcrM on cell proliferation were investigated by conducting a competitive proliferation assay with the cell lines expressing both N6-MTase as WT or mutant compared with untransduced HEK293 cells. Therefore, the fraction of GFP/dsRed double-positive cells relative to non-fluorescent cells was determined. As observed before, the co-expression of the inactive mutants did not compromise the proliferation rate considerably (Fig. 2c). Moreover, the cell lines co-expressing WT CcrM or WT EcoDam together with the inactive mutant of the other N6-MTase showed similar results as observed in the previous proliferation assays. Interestingly, co-expression of WT EcoDam and WT CcrM led to a very minor increase in effects when compared with the expression of CcrM alone (or CcrM together with the inactive mutant of EcoDam). Overall, these data clearly indicate that the gain in methylation at GANTC sequences has a substantially higher impact on the proliferation of HEK293 cells than methylation of GATC sequences, although the overall adenine methylation levels of the genomic DNA were similar after methylation of both motifs.

**RNA-seq analysis after genome-wide adenine methylation by CcrM.** The competitive proliferation assays clearly indicated an impact of CcrM-mediated m6dA methylation within GANTC sequences on cell proliferation. We hypothesized that some of these sequences might overlap with TF binding-sites or other gene regulatory regions, and the adenine methylation might promote or impair the binding of the corresponding factors. For this reason, RNA-seq experiments were conducted with cells containing the WT CcrM expression construct before and after 10 days of dox induction. In addition, CcrM D39A cells were treated identically to control for gene expression changes caused by the dox treatment or the expression of the CcrM protein itself. All RNA-seq experiments were conducted in duplicates. Strikingly, a set of 183 differentially expressed (DE) genes was identified after 10 days of WT CcrM expression using DeSeq2 when comparing with no dox control cells (Fig. 3a). Similarly, 161 DE genes were detected when comparing WT CcrM expressing cells with CcrM D39A expressing cells (Fig. 3b). Importantly, a total of 99 DE genes (66 up-, 33 downregulated) were present in both gene sets (Fig. 3d, Suppl. Table 1), whose expression change in

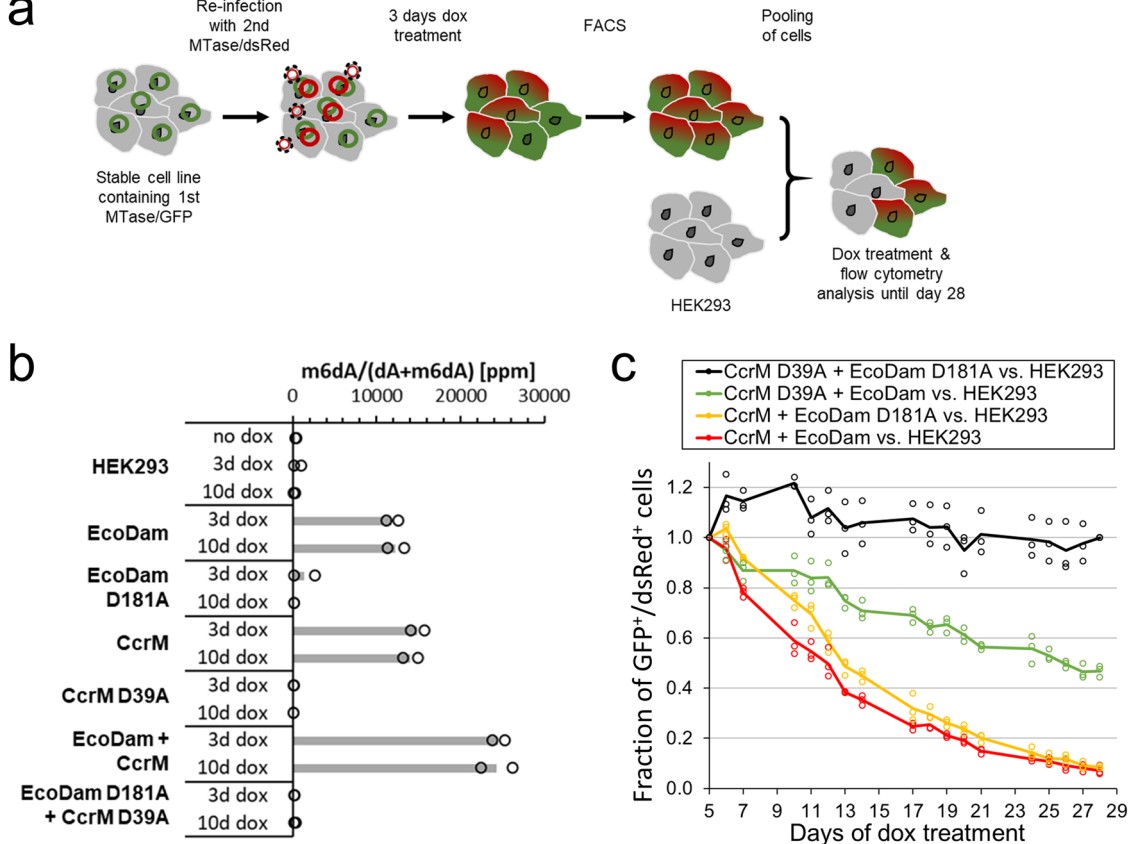

**Fig. 2 Co-expression of two N6-MTases leads to increased m6dA levels in the cells. a** Schematic depiction of the experimental workflow used for the generation of stable double cell lines and the competitive proliferation assay displayed in panel (**c**). **b** Quantification of genome-wide m6dA levels relative to the sum of m6dA and dA in parts per million (ppm) by HPLC-MS/MS measurement. Levels were determined for genomic samples with and without expression of different N6-MTases. **c** Flow cytometry results of the competitive proliferation assay for the cell lines co-expressing two N6-MTases (GFP$^+$/dsRed$^+$) versus non-fluorescent HEK293 cells. The fraction of GFP$^+$/dsRed$^+$ cells was calculated and normalized to day 5 of dox treatment.

turn can be attributed to the introduced m6dA mark rather than to any kind of experimental artifact. The comparison between untreated ('no dox') WT CcrM and dox-treated CcrM D39A cells confirmed the low impact of dox treatment and presence of the CcrM protein on the transcriptome, as only two genes were found to be differentially expressed (Fig. 3c).

**Expression analysis of reporter genes with DE gene promoters.**
We next focused on the promoters of the DE genes in order to analyze whether the transcriptional changes were originating from the methylation changes within these promoters. The borders of the gene promoters were selected based on UCSC browser tracks provided by the Eukaryotic Promoter Database (EPD, https://epd.epfl.ch//index.php)[37], as well as H3K4me3- and H3K27ac ChIP-seq data tracks showing the active promoter regions in HEK293 cells[38]. All promoter regions contained at least one GANTC site (average 7.1 ± 3.2 SD). In the end, the promoters of *EGF*, *ECM1*, *FAM46B* (aka *TENT5B*), *IL6R*, *COL12A1*, *CRABP2*, and *EMILIN2* were chosen for further analysis, because of their consistent and large expression changes after CcrM expression (Suppl. Fig. 4). The promoters were cloned in front of a GFP reporter gene, and the stable WT CcrM or D39A CcrM cell lines (both containing dsRed as reporter) were re-transduced with the newly cloned reporter constructs (Fig. 4a). After antibiotic selection and cell expansion, the median GFP signal of the cell lines was determined by flow cytometry, showing that the respective promoter activities were highly similar

between WT CcrM and D39A cells before dox treatment (Fig. 4b). At this point, the stable cell lines containing the *IL6R* construct were excluded from further experiments due to their very low promoter activity.

Subsequently, CcrM WT and D39A expression was induced by dox treatment for up to one week and the median GFP signal was determined from the dsRed$^+$ cells. By normalizing the GFP signal of CcrM WT cells to that of CcrM D39A cells, the change in reporter gene expression could be entirely attributed to the introduction of genomic m6dA in the cells. Strikingly, four out of the six tested promoters displayed a strong response to the m6dA deposition by CcrM, ranging from 52% (*EGF*) to 86% (*EMILIN2*) reduction in GFP levels (Fig. 4c, Suppl. Fig. 5). Thus, the silencing effect of the m6dA deposition observed in the RNA-seq experiments could be reproduced in this artificial reporter gene setting. It has to be noted that the high promoter activity of *CRABP2* and *COL12A1* observed in RNA-seq (Suppl. Fig. 4b) could not be reconstituted, which suggests that some critical gene-controlling elements (like enhancers) were not included in our constructs. This may also explain the missing effect of CcrM expression on *CRABP2* and *COL12A1* expression.

Next, we were interested to find out whether the m6dA-mediated silencing was stable over longer periods of time. Therefore, the cells containing the *EMILIN2* promoter constructs were first treated with dox for 10 days, and then split and further propagated in medium with or without dox. While the cells constantly treated with dox displayed pervasive silencing of the *EMILIN2* promoter, the cells grown in the absence of dox showed

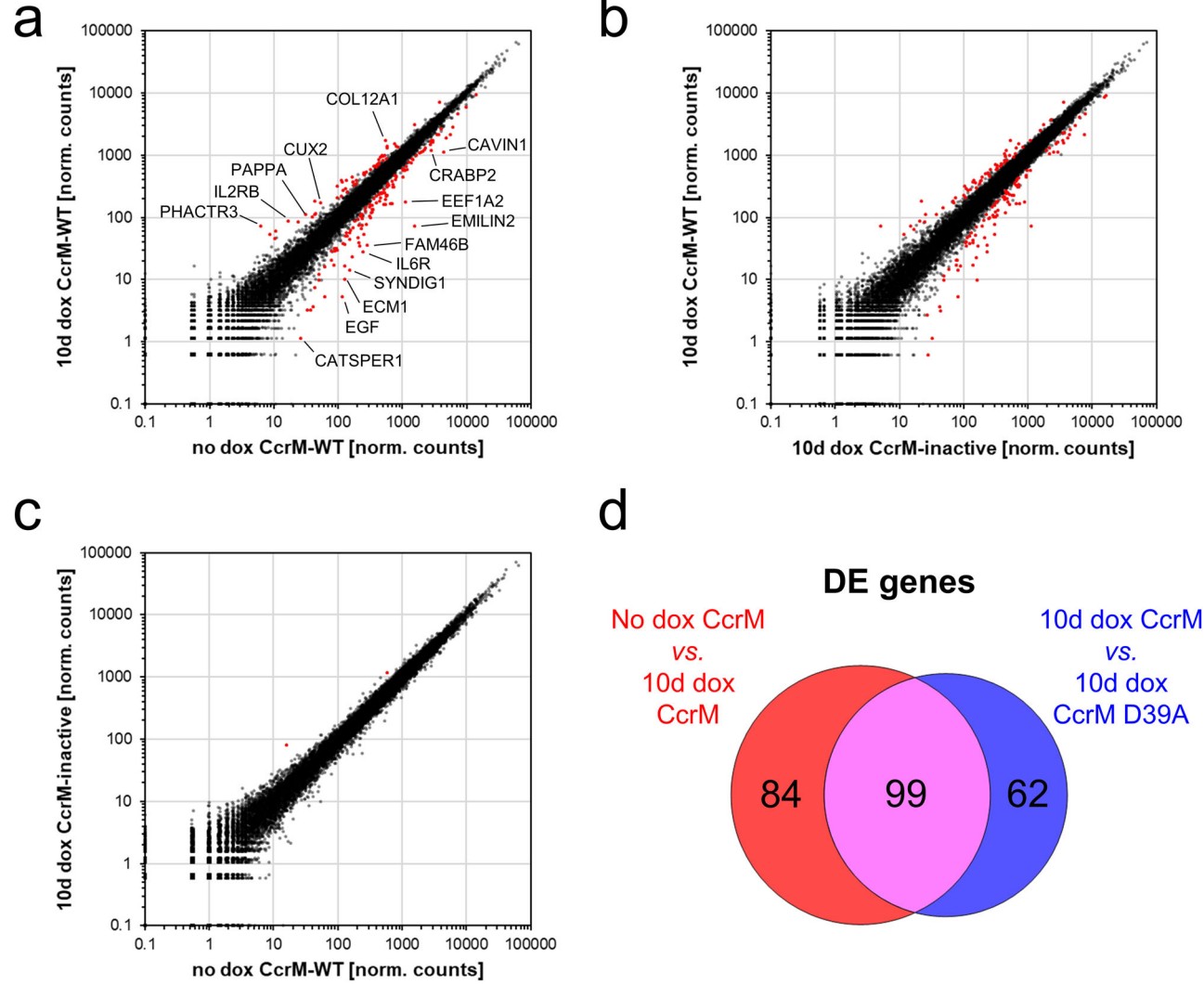

**Fig. 3 Genome-wide deposition of m6dA by the bacterial CcrM MTase leads to the differential expression of multiple genes. a–c** Scatter plots of RNA-seq data normalized by DeSeq2 (normalized counts). Pair-wise comparisons of 'no dox CcrM-WT' versus '10d dox CcrM-WT' (panel **a**), '10d dox CcrM-inactive' (D39A) versus '10d dox CcrM-WT' (panel **b**), and 'no dox CcrM-WT' versus '10d dox CcrM-inactive' (D39A) (panel **c**). Red dots represent differentially expressed genes identified by DeSeq2 ($p < 0.05$ after Benjamini–Hochberg $p$-value correction for multiple testing). A pseudo-count of 0.1 was added to all values. **d** Venn diagram showing the overlap of differentially expressed (DE) genes after 10 days of CcrM expression in comparison to 'no dox CcrM', and '10d dox CcrM D39A'.

a gradual recovery in *EMILIN2* promoter activity (Fig. 4d). This result further highlights the sensitivity of the exemplarily chosen *EMILIN2* promoter towards the m6dA modification and it indicates that the m6dA deposition was transient.

**Upregulated genes show m6dA-dependent reduction of H3K27me3.** In order to unravel the potential mechanisms leading to the transcriptional changes after m6dA introduction at GANTC sites, we analyzed the RNA-seq data in more detail. The DE genes were further investigated based on the direction of their expression change and split into up- and downregulated genes. Both gene sets were analyzed using the Enrichr Tool[39,40] for gene set enrichment. For the downregulated genes, one would either anticipate the m6dA-specific recruitment of a silencing factor, or the loss of DNA-binding of an activating factor (and vice versa for the upregulated genes). Our analysis showed that the upregulated genes were most strongly associated with three different tracks of SUZ12 (ChIP-seq enrichment combined score < 0.003) (Fig. 5a). SUZ12 is a subunit of the Polycomb Repressive

Complex 2 (PRC2), which plays a central role in gene silencing and heterochromatin formation. It is tempting to speculate that the deposited m6dA may reduce PRC2 binding or activity, similarly as PRC2 has been shown to be regulated by CpG sites and m5dC methylation[41–43]. To explore this model, we first inspected H3K27me3 profiles at the promoters of upregulated genes using available ChIP-seq data for HEK293 cells[38]. Then, four upregulated genes were selected for further analysis and the H3K27me3 density was experimentally determined in HEK293 cells with or without expression of WT CcrM or the inactive CcrM mutant by ChIP-qPCR (Fig. 5b and Suppl. Fig. 6a). As expected, H3K27me3 was detected at all four gene loci. Strikingly, however in three cases a clear reduction of H3K27me3 was observed after induction of CcrM expression, but not after expression of the CcrM mutant. In contrast, no changes were observed at the *HOXA11-AS* and *CCND2-AS1* loci, which did not exhibit m6dA-dependent expression changes and were used as controls (Suppl. Fig. 6b). These data suggest that deposition of m6dA in GANTC motifs can lead to a reduction of PRC2 activity at some target genes, accompanied by a decline in H3K27me3

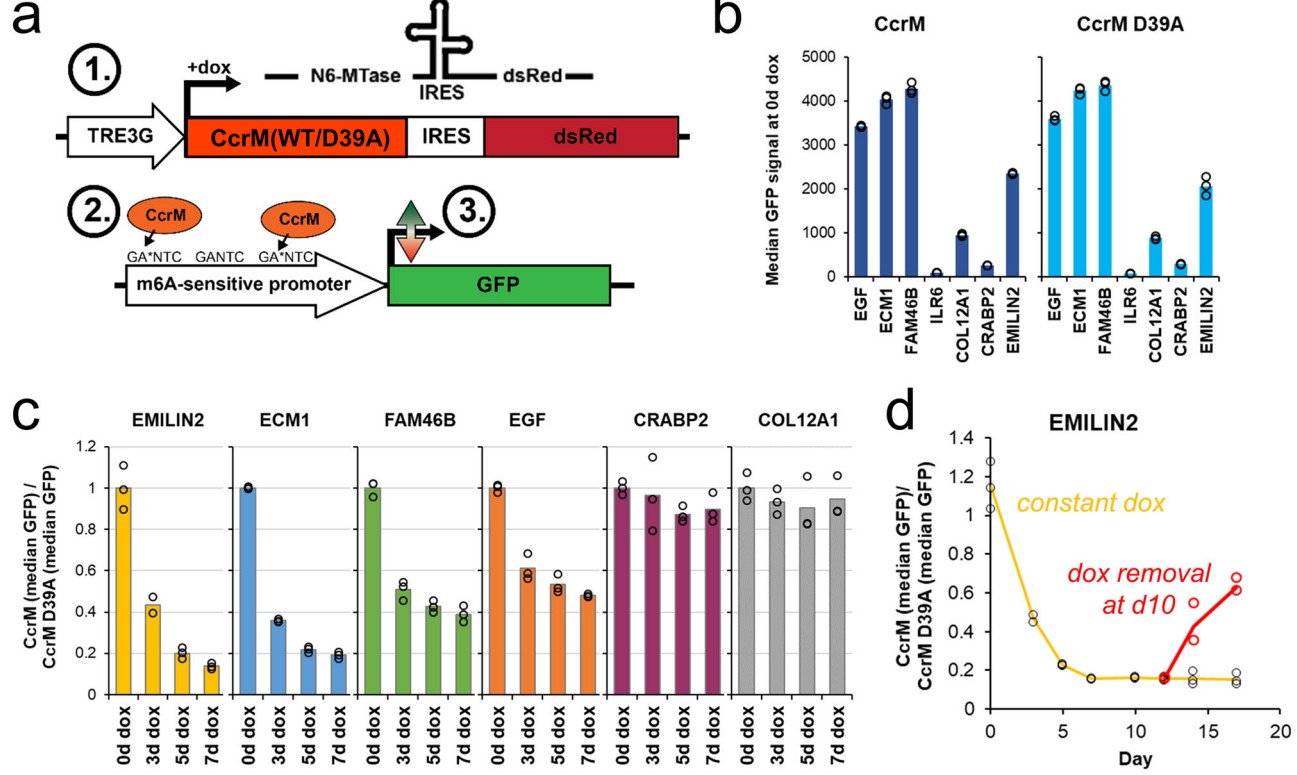

**Fig. 4 Sensitivity of various human promoters to the m6dA modification. a** Expression constructs used to generate stable HEK293 cell lines containing a dox-inducible TRE3G promoter, which allow co-expression of a CcrM variant (WT or D39A) and dsRed over an internal ribosomal entry site (IRES). The same HEK293 cells also contain different versions of m6dA-sensitive promoters upstream of a GFP reporter gene. ① After addition of dox, CcrM is expressed. ② CcrM methylates adenine residues within GANTC motifs of the investigated promoters in an untargeted manner. ③ The m6dA-dependent changes in GFP expression are then tracked by flow cytometry. **b** Median GFP signals of various CcrM (WT or D39A) cell lines expressing GFP under the control of different human promoters. GFP signals were obtained by flow cytometry before dox treatment. **c** Ratio of the median GFP signals of CcrM and CcrM D39A, before and after dox treatment, showing the effects of m6dA on GFP expression. **d** Flow cytometry data showing the dynamics of GFP expression under the control of the m6dA-sensitive *EMILIN2* promoter. Cells were either constantly treated with dox (orange line, black data points), or (starting at day 10) cultivated in the absence of dox (red line and data points).

and de-repression of the silent chromatin state at these genes finally leading to increased gene expression.

**DNA-binding of JUN-family TFs is inhibited by m6dA.** For the downregulated genes, the Enrichr gene set analysis[39,40] revealed a strong enrichment (combined score <0.003) of binding sites of the members of the JUN and FOS TF families (Fig. 5a). Strikingly, all of these TFs contain a GANTC sequence in their core TGANTCA motif, in which the adenine at position 3 is very critical for binding (Fig. 6a). Therefore, it was likely that there is a connection between the gain in adenine methylation and a change in TF binding. For this reason, we analyzed the structure of the FOS/JUN DNA complex (PDB: 1FOS) and modeled two methyl groups to the adenines within the GACTC/GAGTC sequence. A potential steric clash caused by the methyl groups became apparent, because the ω-N of R279 from JUN approaches the C-atom of one of the m6dA methyl groups by 2.3 Å, much closer than the possible van der Waals distance of both atoms (Fig. 6b). Reduction of DNA binding of JUN by the presence of m6dA at position 3 of the TGANTCA binding site was also confirmed experimentally by a DNA binding analysis (Suppl. Fig. 7). Based on the structural similarity, the same mechanism could be applicable for JUND, another member of the JUN family. However, a potential disruption of DNA binding by FOS in the presence of m6dA was not obvious, as no residue of FOS closely approaches the methyl groups introduced at the adenine residues. Based on this, we conclude that the CcrM-mediated

adenine methylation in the core motif of JUN TF family members might impair promoter and enhancer binding of these TFs and thus abolish their positive effects on gene expression.

To test this hypothesis, we re-analyzed the sequences of the m6dA-sensitive promoters used in the cellular experiments. Interestingly, the promoters (*EMILIN2*, *ECM1*, *FAM46B*, *EGF*) which contain at least one complete TGANTCA JUN motif (representing a 5'-T and 3'-A flanked GANTC site) were downregulated most strongly after CcrM induction (Fig. 4c and Suppl. Fig. 5). Almost complete GANTC methylation of the genomic DNA at these m6dA sensitive promoters after CcrM expression was verified by HinfI digestion followed by qPCR (Suppl. Fig. 8). The two other promoters that were not downregulated in the reporter gene assay (*CRABP2*, *COL12A1*) did not contain a TGANTCA sequence in the cloned promoter region, further connecting the effect of adenine methylation at TGANTCA sites and reduction of gene expression. ILR6, which was strongly downregulated in RNA-seq, but did not show good promoter activity in the reporter gene assay, contains a TGANTCA motif in the part of the promoter that was not included in the reporter gene assay (Suppl. Figs. 4 and 5).

For further experimental validation, the *EMILIN2* promoter was selected, because it contains only one complete TGANTCA motif. By mutating this site to TGGNTCA ('mut1'), the binding of the JUN proteins should be abolished, thus simulating the absence of the TF at this site. As a control, the upstream GANTC site, which does not contain the flanking bases required for

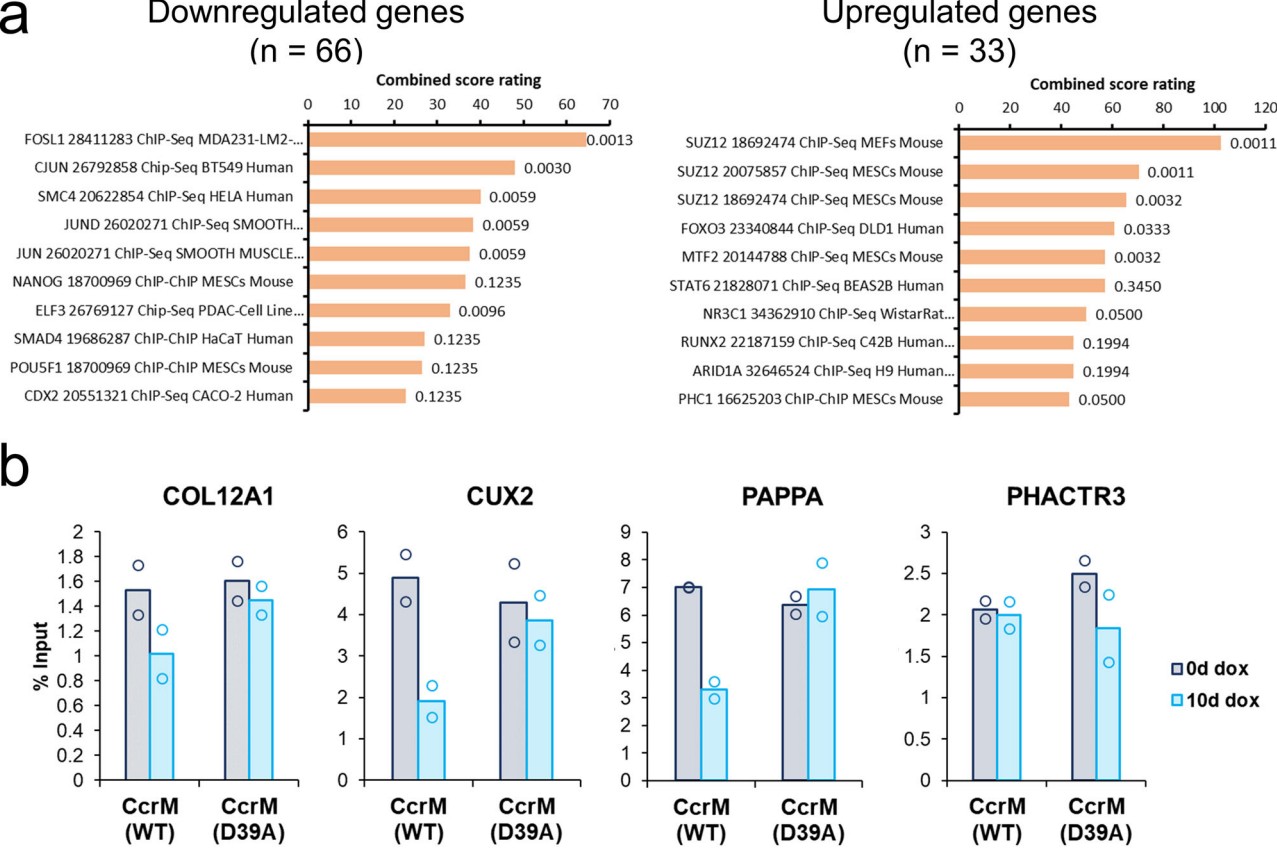

**Fig. 5 Analysis of m6dA-dependent regulated genes. a** Gene set enrichment analysis of significantly down- or upregulated genes utilizing the ChEA 2022 results available through the Enrichr tool (https://maayanlab.cloud/Enrichr/). **b** Change of H3K27me3 in the promoter regions of upregulated genes. Shown are H3K27me3 ChIP-qPCR results in two biological repeats (indicated by the dots), each of them based on three technical repeats of the qPCR. Three out of four tested genes (*COL12A1, CUX2, PAPPA*) showed a CcrM-dependent decline in H3K27me3. The qPCR amplicons, genomic ChIP-seq profiles and qPCR results of control experiments for the *HOXA11-AS* and *CCND2-AS1* loci are shown in Suppl. Fig. 6. The bars indicate the average values of two experimental repeats.

JUN-binding, was also mutated to GGNTC ('mut2'). Strikingly, the loss of the JUN target motif resulted in a 7 to 8-fold reduction of GFP expression compared to the wildtype variant (Fig. 6c), and thereby closely reflected the loss of GFP expression caused by m6dA introduction after 10 days of CcrM expression (Fig. 4c). In contrast, mutating the upstream GANTC sequence did not change expression of the reporter gene. Moreover, *EMILIN2*$^{mut1}$ (−48%) could not be silenced to the same extent as *EMILIN2*$^{WT}$ (−86%) or *EMILIN2*$^{mut2}$(−87%) by CcrM expression (Fig. 6d). In combination, these data show that expression of the *EMILIN2* promoter was dependent on an intact JUN binding motif, while the repression of expression by m6dA was dependent on an intact GANTC motif. These findings document the combined role of JUN and GANTC motifs in the m6dA mediated silencing processes observed in our study for this example promoter and indicate that adenine methylation of a single TGANTCA motif within the *EMILIN2* promoter was primarily responsible for the m6dA-dependent control of *EMILIN2* expression.

## Discussion

During the last years, several publications have reported variable levels of m6dA in the genomic DNA of human cells and connected these findings with important effects in physiology and disease. At the same time, other publications raised doubts about these findings leading to a major uncertainty in the field regarding key observations starting from the mere existence and levels of genomic m6dA in human cells, its way of incorporation

and potential biological effects leading to the putative human N6-MTases (reviewed in[9–11]). Hence, there is an urgent demand for novel, alternative research approaches leading to a set of fundamental results that can be accepted by the field and used as a foundation and starting point for further investigations. Indeed, the levels of m6dA in HEK293 cells observed in our experiments were at the detection limit and many additional technical and biological controls would have been needed to assess if this represents true biological DNA methylation. Given that low levels of endogenous m6dA, as well as the lack of non-controversial bona fide human N6-MTases, are recurrent technical hurdles in the study of potential effects of m6dA in human cells, we decided to employ an orthogonal approach by expressing well-characterized, highly active bacterial N6-MTases in human cells followed by the analysis of the cellular effects of the artificially introduced genomic m6dA. This procedure allowed us to deliver high levels of m6dA into GANTC and GATC motifs in a genome-wide scale and to investigate the biological effects of m6dA in these contexts. Although these high methylation levels differ from endogenous m6dA profiles in human cells determined by different groups[8,12,13,16–20], some of the GANTC and GATC methylation sites may overlap with endogenous motifs. Our data show that genome wide methylation of GANTC or GATC motifs has direct effects on HEK293 cell viability, indicating potential biological effects of endogenous m6dA. However, effects were only moderate with differences in cell viability appearing over several days, indicating that global genome-wide

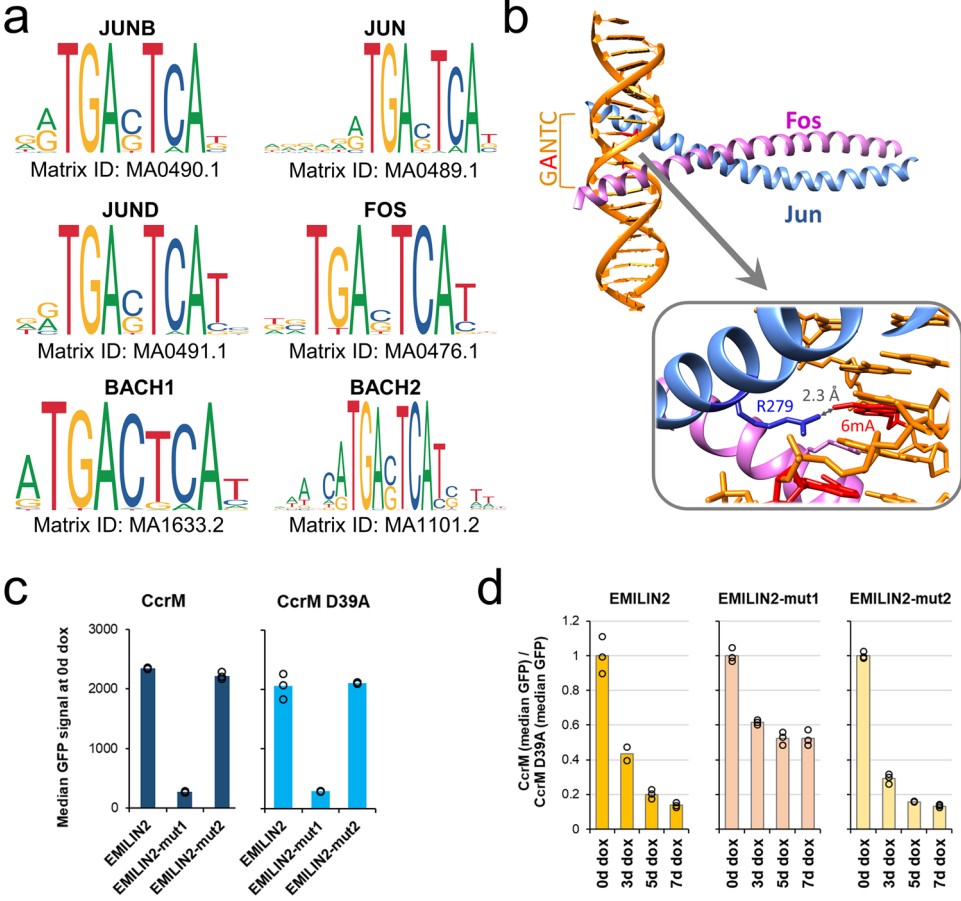

**Fig. 6 m6dA-dependent downregulation of genes is potentially linked to loss of JUN TF binding. a** Position weight matrices (PWM) of candidate TFs containing the CcrM target motif. PWMs were obtained from the JASPAR 2022 database (https://jaspar.genereg.net/). **b** Crystal structure of the Jun/Fos heterodimer (pdb: 1FOS) binding to DNA (orange). Deoxyadenine residues in the GANTC sequence on both strands of the model (red) were modified to m6dA. The distance between the amino group of R279 of JUN and the adjacent m6dA methyl group is indicated. **c** Flow cytometry results, showing the median GFP expression in cells containing different variants of the *EMILIN2* promoter upstream of GFP: unaltered *EMILIN2* promoter, A-to-G mutation in the putative JUN (TGANTCA) motif (mut1), or A-to-G mutation in a GANTC motif (mut2) lacking the surrounding T/A bases. **d** Flow cytometry analysis showing the changes of GFP expression for the different *EMILIN2* promoter variants after CcrM induction. Ratios of the median GFP signals for CcrM and CcrM D39A were calculated.

methylation of GANTC and GATC motifs (even together) is tolerated surprisingly well in human cells. On the other hand, the effects of methylation in GATC motifs on cell viability were lower than those of GANTC methylation, despite similar global levels of m6dA, indicating that the effects of m6dA are specific for each methylation motif. Hence, it is possible that methylation within other motifs could have stronger biological effects in cells.

We next identified genes that are directly regulated by GANTC methylation. Downregulation of genes can be explained by the m6dA dependent inhibition of DNA binding of stimulatory factors, while upregulation of genes can be explained by the inhibition of the activity of repressive factors. Our data show examples for both mechanisms, because adenine-N6 methylation of GANTC motifs can lead to reduction in H3K27me3 levels and hence upregulation of genes that were previously silenced by the Polycomb system. Moreover, we document that m6dA within the GATNTC motif of TGANTCA sites can reduce gene expression by inhibition of DNA binding of JUN family TFs. Using the *EMILIN2* promoter as model system, we have shown that the presence of m6dA at position 3 in single TGANTCA motifs is sufficient to decrease gene expression. Methylation in other motifs than GANTC might affect DNA binding by other TF-families as illustrated by the m6dA mediated reduction of DNA

binding of the SATB1 protein[14]. In the biological context, our data indicate that targeted local introduction of m6dA at individual GANTC sites in PRC2 or JUN binding sites could directly up- or downregulate the expression of associated genes, even if the overall global levels of m6dA were low.

In summary, the strong gene-regulatory potential of m6dA discovered in our study indicates that this DNA modification can have important biological effects under certain physiological or pathophysiological conditions. Our data provide a fundament for further investigation of the biological roles of m6dA in human DNA. The technologies established here will allow the identification of more m6dA-regulated genes, potential m6dA readers and human N6-MTases, finally helping to clarify the role of this DNA modification in human physiology.

## Methods

**Cloning of vectors for stable cell lines**. The full-length constructs of EcoDam variants were amplified from pET28a-EcoDam(WT/D181A) vectors[44], the full-length constructs from CcrM variants from pET-CcrM and pBAD-CcrM(D39A)[45,46]. For this, primers with overhangs containing XhoI (5′ end of the MTase gene) and MluI (3′ end of the MTase gene) restriction sites were designed. Subsequently, the N6-MTases were inserted into the targeting vectors (TRE3G-DNMT3A(CD)-IRES-GFP-PGK-NeoR or TRE3G-ZnF-DNMT3A(CD)-IRES-dsRed-PGK-HygroR[38]) either by restriction ligation cloning or by Gibson

Assembly. The m6dA-sensitive promoters selected for GFP reporter gene assays were amplified from HEK293 genomic DNA. The obtained amplicons were cloned into the TRE3G-GFP-PGK-NeoR vector by restriction ligation cloning, whereby the TRE3G promoter was replaced. Subsequently, the stable HEK293 cell lines already containing TRE3G-CcrM(WT/D39A)-IRES-dsRed-PGK-HygroR were re-infected as described below (in the section "Generation of stable HEK293 cell lines"), followed by antibiotic selection and recovery. The integrity of all cloned vectors was validated by Sanger sequencing.

**Cell maintenance.** Human Embryonic Kidney 293 cells expressing the murine ecotropic retroviral receptor (EcoR)[38] (HEK293) and Platinum-E (PlatE) cells were cultivated in DMEM + medium (DMEM supplemented with 10% Fetal Bovine Serum, 1% penicillin/streptomycin, 4 mM L-glutamine) in an incubator (BINDER) at 37 °C, 95% relative humidity and 5% $CO_2$. The cells were split every 2–3 days in a ratio between 1:4 and 1:8 to keep confluency below 90%. The splitting was performed by washing the cells with PBS (w/o $Ca^{2+}$ and $Mg^{2+}$) pre-warmed to 37 °C and by immersing the cells with a Trypsin–EDTA solution (Sigma) for 5 min at 37 °C. Afterwards, the cells were resuspended in fresh DMEM+ to stop the trypsinization and re-seeded in the desired dilution and ratio. To exclude mycoplasma contamination, the cells were routinely stained with DAPI and absence of mycoplasma DNA was shown by qPCR.

**Generation of stable HEK293 cell lines.** Stable HEK293 cell lines were generated as previously described[38]. In brief, MSCV retrovirus particles were produced by co-transfecting PlatE cells with the cloned vector and a GagPol helper plasmid using calcium phosphate transfection. Forty-eight hours after transfection, the supernatant containing the retrovirus particles was harvested and filtered through a 0.45 µm syringe filter. Next, $3 \times 10^5$ HEK293 target cells (expressing EcoR) were seeded into the well of a six-well plate, and the virus supernatant was added dropwise. Forty-eight hours post infection, antibiotic selection (300 µg/mL neomycin or 200 µg/mL hygromycin) was initiated. After the selection process, the infected cells were expanded, and either used directly for cell culture experiments, or cryopreserved at -80 °C upon later usage. For the simultaneous expression of two N6-MTases and in the GFP-reporter assay, stable double cell lines were generated. For this, the previously established single cells lines were re-infected with a second construct using the same workflow as described above. In each case, the infected and selected cells were transferred into a T-75 flask (Sarstedt) and expanded. A first fluorescence activated cell sorting (FACS) step was performed as described in the next paragraph in order to remove "leaky" cells expressing the constructs without doxycycline (dox) treatment.

**Cell sorting and flow cytometry.** For detaching, the cells were washed with 8 ml PBS (w/o $Ca^{2+}$ and $Mg^{2+}$), then incubated for 5 min with 1.5 mL Trypsin-EDTA solution (Sigma) and subsequently resuspended in 8.5 mL DMEM + . The cells were centrifuged in a 15 mL reaction tube for 5 min at 300 rcf, then the supernatant was removed, and the cells were resuspended in 1 ml DMEM + . To remove any aggregates or large particles, the cells were filtered through a 70 µm separation filter (Miltenyi Biotec) and collected in a 15 ml reaction tube. To collect the sorted cells, the inner surface of an empty 15 mL reaction tube was covered with 2 mL DMEM + and pre-cooled on ice. The cells were sorted using the Cell Sorter SH800S (Sony Biotechnology). Sorting was conducted for single cells combined with gating with respect to the fluorescence in GFP and RFP channels as described in the text (see Suppl. Figs. 9 and 10 for gating strategies). Spillover of signals originating from different fluorophores was corrected by compensation. Subsequently, the sorted cells were centrifuged for 5 min at 300 rcf and cultured in a T-75 flask for genomic DNA isolation.

**Competitive proliferation assay.** To investigate the effects of m6dA on cell proliferation and viability, competitive growth assays were conducted. In these experiments, stable HEK293 cells expressing the catalytically active MTases were co-cultivated with HEK293 cells that were either uninfected or expressed the respective inactive mutant of the MTase. One week after viral transduction followed by selection, the experiments were started by treating the stable HEK293 cell lines constantly with 1 µg/mL dox to induce expression of the DNA MTases.

As a first experimental setup, MTases were co-expressed with a green fluorophore in HEK293 cells. After three days of dox treatment, the infected cells were enriched by FACS and mixed 1:1 with untreated sorted HEK293 cells using $5 \times 10^5$ cells of each type. The pooled cells were divided into three wells of a 24-well plate each and evaluated as replicates. The cells were cultured, passaged every 24 h according to their confluency and constantly treated with 1 µg/mL dox. During passaging, the cells were resuspended in a total volume of 1 mL in DMEM + and 150 µL of this cell suspension were transferred to a 96-well plate for flow cytometry measurement using the MACS Quant VYB (Miltenyi Biotec). The GFP fluorescence of the cells was measured every weekday for 28 days. The data was analyzed using the FlowJoR V.10.5.13 (FlowJo LLC) software and living single cells were divided into GFP-positive (GFP+) and GFP-negative (GFP-) populations to determine the ratio of cells carrying the DNA MTase and untreated cells.

In a second experimental setup, two different MTases co-expressing fluorophores of different color and selection markers were co-cultivated within the same vessel, i.e., the active MTase co-expressing GFP and the inactive MTase co-expressing dsRed. After three days of dox treatment, infected cells were isolated by FACS as described above and pooled 1:1 using $5 \times 10^5$ cells of each type. Afterwards, cells were treated as described above. Living single cells were divided into GFP+ and dsRed+ populations to determine the ratio of cells carrying the active DNA MTase and cells carrying the inactive DNA MTase.

In a third experimental setup, two different active MTases co-expressing fluorophores of different color and selection markers were co-expressed within the same cell. By this approach, the amount of m6dA in the cells could be further increased by targeting of two independent methylation motifs. After three days of dox treatment, the infected cells expressing two MTases were enriched as described above and pooled 1:1 with untreated HEK293 cells using $5 \times 10^5$ cells of each type. Afterwards, the cells were treated as described above. Living single cells were divided into GFP+/dsRed+ and GFP−/dsRed− populations to determine the ratio of cells expressing MTases and untreated cells.

**GFP reporter assay to test m6dA-sensitive promoters.** GFP reporter assays were conducted in biological triplicates. For this, $10^5$ cells per replicate were seeded into the well of a 24-well plate. Before dox treatment, the GFP fluorescence signal was measured by flow cytometry (MACS Quant VYB, Miltenyi Biotec) gating on all living single cells. The co-expression of CcrM and dsRed was constantly induced throughout the entire time course experiment by supplementing the cell culture medium with 1 µg/mL dox. Every two to three days, the median GFP fluorescence signal was determined in all living single cells showing red fluorescence.

**Genomic DNA digestion.** Genomic DNA (gDNA) was extracted from parental HEK293 cells or from cells that expressed different variants of m6dA MTases for up to 10 days. Frozen cell pellets were thawed on ice and resuspended in 200 µL PBS (w/o $Ca^{2+}$ and $Mg^{2+}$). The gDNA was isolated using the QIAamp DNA Mini Kit (Qiagen) according to the manufacturer's instructions and eluted with 200 µL ddH$_2$O pre-warmed to 70 °C. The concentration of the isolated gDNA was determined based on the absorbance at 260 nm as measured by NanoDrop 1000 (Thermo Fisher Scientific). Afterwards, gDNA (250–500 ng) was digested separately with 0.5 U of the restriction endonucleases DpnI (New England Biolabs), DpnII (New England Biolabs) or HinfI (New England Biolabs) at 37 °C overnight using the corresponding reaction buffers. The digested products were analyzed on a 1% agarose gel stained by GelRed (Genaxxon).

**m6dA sensitive qPCR.** Genomic DNA from CcrM WT expressing cells was isolated and digested with HinfI as described in the previous section. For m6dA sensitive qPCR, amplicons were designed in the m6dA sensitive promoter regions identified in our work (EMILIN2, ECM1, FAM46B, EGF (Suppl. Fig. 5 and Suppl. Table 2). The presence of m6dA at the CcrM target motif blocks cleavage by HinfI, leaving an intact qPCR template. qPCR was conducted basically as described[38], using undigested and digested genomic DNA (5–20 ng) from three biological samples (no dox, 3d dox, 10d dox) as template. The starting quantities were calculated using a standard curve. Variations in DNA amounts between samples were normalized to a control region in the EMILIN2 promoter, which does not contain HinfI restriction sites. Subsequently, undigested and digested no dox and 3d dox samples were normalized to the respective 10d dox sample.

**HPLC–MS/MS.** About 1 µg gDNA was fragmented using the EpiShear Probe Sonicator (Active Motive). For sonication, the sample volume was adjusted to 200 µL and a probe with a tip diameter of 2 mm was used. Each sample was treated with 20 pulses of 20 s with an amplitude of 25% and a pause of 30 s after every pulse. The samples were purified using the NucleoSpin Gel and PCR Clean-up Kit (Macherey-Nagel), eluted in 50 µL water and the concentration was measured with NanoDrop 1000 (Thermo Fisher Scientific). Afterwards, an aliquot of 35 µL of the sonicated genomic DNA was digested to nucleosides in two steps following the digestion and filtration protocol as described[47]. The following external standards were used: unmodified dATP, dGTP, dCTP, and dTTP were from Genaxxon Bioscience, m6dATP was from TriLink BioTechnologies (#N-1083). External standards were digested and treated as described above for the gDNA samples.

The digested sample and standard solutions were then further diluted with a mixture of 1% acetonitrile and 0.1% formic acid in a 1:1 ratio and 20 µL were injected into an Impact II Quadrupol-Time-of-Flight (qToF) mass spectrometer (Bruker Daltonik) coupled to an Infinity II HPLC system (Agilent). LC separation of DNA samples was carried out on a $3 \times 150$ mm, 3 µm PurospherR STAR RP-8 endcapped column (Merck) at a flow rate of 225 µL/min. The column oven temperature was set to 45 °C, and a binary gradient of water/acetonitrile (ACN), containing each 0.1% formic acid, was used for separation: 3% ACN for 5 min, 3–100% in 9 min, 100% for 3.9 min, back to 3% in 0.1 min, followed by re-equilibration for 7 min. Samples were stored in the autosampler at 10 °C.

For MS-analysis, the mass spectrometer was run in positive full-scan as well as multiple reaction-monitoring (MRM) mode. Mass range was set to $m/z$ 50–450 and mass spectra were acquired by time-of-flight ($r = 35,000$ @ $m/z$ 300) with a spectra rate of 2 Hz. Electrospray ionization was applied, and the source parameters were: capillary voltage of 4500 V, nebulizer gas pressure of 2.0 bar, dry gas flow rate of 8.0 l/min and dry gas temperature of 250 °C. Mass accuracy was assured by internal

calibration of each measurement by 10 mM sodium formate calibration solution in water/*n*-propanol (1:1, v/v), that was injected with a flow rate of 180 μL/min.

For MRM analyses, precursor masses were separated with a width of 1 Dalton, and collision energies were set to 20 eV. All DNA compounds were analyzed as $[M + H]^+$ ions. HPLC–MS/MS analyses were run by Compass HyStar v4.1.21.2 and oToF Control v4.1 (Bruker Daltonik). Further data evaluation was performed using the Skyline (64-bit) v19.1.0.193 software. Data analysis was carried out under the assistance of Skyline by calculation of the peak area of the transitions of precursor and product ions set as follows: dA *m/z* 252.1097 to 136.0623 and m6dA *m/z* 266.1253 to 150.0780, respectively (Suppl. Fig. 3c). Compound identification was additionally confirmed by reproducible retention times of 6.5 min for dA and 12.43 min for m6dA for the measured standards and samples (Suppl. Fig. 3a). Calibration curves were established using external dA and m6dA standards (Suppl. Fig. 3b).

**Library generation for RNA-seq.** Total RNA was isolated from frozen cell pellets containing ~$10^6$ cells using the RNeasy® Mini Kit (Qiagen) and was then quantified via NanoDrop1000. For each RNA-seq library, 1 μg RNA was taken as input material. Since total RNA mostly constitutes of rRNA (≥ 80%), poly-adenylated mRNA was first enriched using the NEBNext® Poly(A) mRNA Magnetic Isolation Module (NEB). Afterwards, RNA-seq libraries were generated using the NEBNext® Ultra™ II Directional RNA Library Prep Kit for Illumina® (NEB), in combination with NEBNext® Multiplex Oligos for Illumina® (Index Primers Set 1) (NEB), thus resulting in strand-specific RNA libraries (opposite sense strand). All library preparation steps were conducted according to the manufacturer's instructions. For quality assessment of the RNA-seq libraries, fragment size distributions and concentrations were determined by Labchip® GX Touch (Perkin Elmer). Finally, 300 ng of each RNA library was submitted to Novogene Corporation (UK) for Next-Generation-Sequencing (Illumina Novaseq 6000 system, paired-end, 2× 150 bp-mode). Two independent experiments were conducted for each condition.

**RNA-seq data processing.** Sequencing data were received in FASTQ format (2× ~20 million reads/sample), and all subsequent data processing steps were performed on the European Galaxy Platform (usegalaxy.eu[48]). The quality of the reads was analyzed using the FASTQC tool, followed by trimming of Illumina adapters using the Trimmomatic tool using the options: ILLUMINA-CLIP step; TrueSeq3 paired-end for MiSeq and HiSeq[49]. Paired-end reads were mapped on the human reference genome Hg38 by utilizing the gapped-read mapper RNA STAR in default settings[50]. The obtained BAM files, as well as the refFLAT transcript file (GTF format), served as input to generate count tables using the HTseq-count tool in 'reverse strand' mode[51]. Finally, differential gene expression testing was conducted by employing DeSeq2[51], using the previously obtained HTseq-count tables by combining the two corresponding replicates. DeSeq2 utilizes the median-of-ratios method for normalization. The resulting normalized counts file was taken as a basis for data depiction. Differentially expressed genes were called on the basis of $p < 0.05$ after Benjamini & Hochberg *p*-value correction for multiple testing. Gene set enrichment analysis was conducted using the Enrichr Tool with default settings. The obtained ChEA (ChIP-X Enrichment Analysis) 2022 tables were exported from the website and sorted by the combined score (https://maayanlab.cloud/Enrichr/)[39,40].

**JUN overexpression, purification and gel shift assay.** The JUN expression construct pET-29b-3xGluGlu-JUN-His6 was a gift from Kevin Janes (Addgene plasmid # 129227)[52]. Expression and purification were conducted basically as described[52] with some modifications. For protein expression, the expression plasmid was transformed into BL21 (DE3) Codon+ RIL *E. coli* cells (Stratagene). A single colony was used to inoculate a 30 mL preculture in LB medium, which was grown at 37 °C for 7 h. Subsequently, 1 L main expression culture was inoculated with 10 mL preculture and grown at 37 °C until reaching an OD(600 nm) of 0.4. At this point, protein expression was induced by addition of 100 μM IPTG and cells were grown at 16 °C for 12 h. Then, cells were harvested by centrifugation at 4 °C and 5000 rcf for 15 min, washed with STE buffer (10 mM Tris/HCl pH 8, 100 mM NaCl, 1 mM EDTA), pelleted again and resuspended in sonication buffer (54 mM Tris/HCl pH 7.4, 160 mM NaCl, 1 mM EDTA, 1 mM DTT, 10 mM MgSO₄, 13 mM MgCl₂). Cells were disrupted by sonication for 10 min (40 cycles, 15 s active sonication, 30 s pause, 90% amplitude) using the EpiShear Probe Sonicator (Active Motif). The cell lysate was centrifuged at 47,000 rcf for 60 min. The supernatant was filtered through a 0.45 μm syringe filter (Chromafil GF/PET 45, Macherey-Nagel) and passed over a column containing 1 mL Ni-NTA Superflow beads (Qiagen), which was equilibrated with 20 mL sonication buffer, using a NGC Quest Plus FPLC system (BioRad). Unbound proteins were washed from the column with 30 mL sonication buffer and subsequently with 40 mL wash buffer (25 mM sodium phosphate pH 7.2, 250 mM NaCl, 1 mM EDTA, 0.2 mM DTT, 20 mM imidazole). The protein was eluted with 15 mL elution buffer (25 mM sodium phosphate pH 7.2, 250 mM NaCl, 1 mM EDTA, 0.2 mM DTT, 220 mM imidazole) and fractions of 500 μL were collected. The fractions with the highest protein yield were pooled and dialyzed for 3 h against dialysis buffer (25 mM sodium phosphate, 160 mM NaCl, 1 mM EDTA, 1 mM DTT, 10% glycerol). Aliquots were snap frozen in liquid nitrogen and stored at −80 °C. The protein quality was confirmed via SDS-PAGE stained with Coomassie BB and the concentration was determined by Nano-Drop1000 (Thermo Fisher Scientific).

The DNA binding assays were conducted with 37mer Cy5-labeled double-stranded DNA substrates containing the JUN target sequence TGAGTCA with a methylated or unmethylated adenine in both DNA strands of the central GANTC motif (Suppl. Table 2). The upper DNA strand also carried a Cy5 label on the 5′ end for detection. For the gel shift DNA binding assay, 50 nM of the Cy5-labeled double-stranded 37mers were incubated for 1 h at 37 °C with different amounts of the purified JUN protein in binding buffer (20 mM Tris/HCl pH 7.4, 40 mM KCl, 5 mM MgCl₂, 5% glycerol) in a total volume of 10 μL. During the incubation, a 6% polyacrylamide gel was pre-run at 200 V for 40 min at 8 °C in 0.25× TBE Buffer (22 mM Tris-HCl pH 8.0, 22 mM boric acid, 0.5 mM EDTA). Afterwards, 5 μL of the samples were loaded and electrophoresis was performed at 70 V for 90 min. The Cy5-labeled DNA was detected with UV light using a Fusion advance solo 4 (PeqLab) with a F-695Y interference filter. The GeneRuler DNA ladder mix (ThermoFisher Scientific) was stained with GelRed (Genaxxon).

**H3K27me3 ChIP-qPCR.** The H3K27me3 ChIP analysis was conducted basically as described[38] using HEK293 cells containing WT or mutant CcrM with or without dox induction. Dox-induced cells were FACS sorted and finally $1 \times 10^6$ cells were used in each case. For IP, the H3K27me3 antibody (#39155, Active Motif) was used (5 μg per IP). The qPCR analysis was conducted and analyzed as described[38] using the qPCR primers specified in Suppl. Table 2.

**Statistics and reproducibility.** The number of independent experimental repeats is indicated for each experiment. Cell proliferation data were based on three to six independent repeats. HPLC/MS and RNA-seq were based on two independent repeats, qPCR on two to three independent repeats. *p*-Values were determined by DeSeq2 using the median-of-ratios method for normalization including Benjamini–Hochberg *p*-value correction for multiple testing.

**Reporting summary.** Further information on research design is available in the Nature Portfolio Reporting Summary linked to this article.

## Data availability
The RNA-seq data are available at DaRUS under https://doi.org/10.18419/darus-2813 (ref. [53]). Source data and uncropped images are provided with in Supplementary Data 1. All other data are available from the corresponding author on reasonable request.

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

## Acknowledgements

This work was partially supported by the University of Stuttgart through the Terra Incognita program and the DFG (JE 252/35-1).

## Author contributions

Ju.B. and F.K. conducted all biochemical and cell culture experiments with help of A.K. F.K. and B.O. conducted the HPLC–MS measurements with support from S.A. T.C. conducted the H3K27me3 ChIP analysis. A.K. conducted the JUN DNA binding experiments. Je.B. supervised the HPLC–MS measurements and corresponding data analysis. Ju.B. and A.J. prepared the paper draft. Ju.B., A.K., T.C., F.K., and A.J. prepared the figures. A.J. devised and supervised the work. All authors contributed to data interpretation and editing of the paper. The final paper was approved by all authors.

## Funding

## Competing interests

The authors declare no competing interests.
