## [Peer Review File · Communications Biology]

Reviewers' comments:

Reviewer #1 (Remarks to the Author):

The article from Broche et al. aims at describing functional relationships between gene expression and 6mdA deposition by bacterial DNA methyltransferases (EcoDam and CcrM) in H293 cells. The authors first successfully validate the expression of wild-type and mutant DNA methyltransferases as well as 6mdA deposition and identify a decrease of proliferation upon expression of these enzymes, linked to deposition of 6mdA. They further identify a very small set of genes (less than 100 genes) to be differentially expressed upon expression of wt CcrM. Amongst these, 7 of them are enriched with GANTC motifs (recognized by CcrM) and are expressed in a luciferase assay, of which 6 expressed at sufficient levels. The authors observed that 4 of those 6 genes are directly regulated by CcrM. Next, they aimed at finding mechanistic links between 6mdA deposition and gene expression, and found a handful of genes regulated by PCR2/H3K27me3 as well as with JUN family of transcription factors. Overall, I find the article to be of moderate interest, but acknowledge the need for understanding the dynamics of gene expression and 6mdA deposition in vivo.

My major concern is that H293 cells lack 6mdA (at least as identified by restriction assays and MS/MS in this manuscript). I initially thought that overexpressing bacterial DNA methyltransferases would help identify a subset of genes that would be regulated by 6mdA deposition, and that the authors would then quest for looking at endogenous 6mdA on these targets (for example by immunoprecipitation with an antibody raised against 6mA), but surprisingly it was not the case. Instead, they focused on examining regulation mechanisms on a very low number of targets. The purpose of studying such an artificial system is to link it to endogenous m6dA deposition, otherwise it seems to me that the authors are focusing on an artifact rather than true biology. Further, it is expected to have a handful of targets (99) differentially expressed upon deposition of an exogenous epigenetic mark, of which it is also not surprising that a few are PRC2 or JUN -regulated. In brief, unless Prof. Jeltsch's team can find natural m6dA deposited on a significant subset of the 99 deregulated targets, I unfortunately believe that the manuscript is not suitable for publication in Communications Biology.

Reviewer #2 (Remarks to the Author):

The presence and functions of DNA 6-methyladenine (m6dA) have been heavily debated in the recent years, and several studies concluded m6dA was purely artifact in the genome, due to potential bacteria contamination. However, others insisted the presence of m6dA in metazoans, despite in very low amount due to active or passive mechanisms, and their potential epigenetic functions. In this study, Broche et al took the biorthogonal approach to overexpress two well-established bacterial N6-MTases, EcoDam and CcrM, to monitor the roles of high m6dA deposition in human HEK293 cells genome. High level of m6dA, validated by m6dA sensitive restriction enzyme digestion and quantitative HPLC, significantly impair normal cell proliferation. Global m6dA alteration resulted in a group of significantly dysregulated genes that were potentially modulated by promoter m6dA deposition. M6dA could crosstalk with Polycomb/H3K27me3 and DNA binding JUN-family transcription factors to modulate gene expression in both endogenous loci and trans-genes. Overall, this is a well-tailored studies addressing a critical and confusing question – does m6dA indeed epigenetically regulate mammalian gene expression. One cannot ignore potential biological roles of m6dA simply because it is sparse in the genome. This study effectively contributes to our understanding of m6dA in mammalian genome, which could be transiently or loci-specifically enriched to a meaningful amount to exert epigenetic roles. However, several conceptual and technical gaps should be resolved before moving forward with this manuscript:

1. The authors designed N6-MTase expression constructs to separately express N6-MTase and GFP under DOX control, then mixed the GFP positive cells with parental 293 cells to monitor and compare their growth. However, GFP expression is under the control of IRES, which might not necessarily

synchronize with the N6MTase expression. The enrichment of m6dA from GFP positive cells, comparing to no GFP cells, should be carefully tested. The co-expression of N6-MTase and GFP should be monitored by immunofluorescence.

2. While EcoDam and CcrM expression both drive the m6dA production, it is plausible they regulate different subset of m6dA deposition to specific loci. Particularly, CcrM displayed much more strong cellular effects in HEK293 cell proliferation and was likely to largely responsible for the synergistic effects of EcoDam and CcrM co-expression (Fig. 2C). It will be much more informative if the authors could provide genome-wide m6dA profile, or at least validate m6dA on several key loci to delineate loci-specific roles of these methyltransferases.

3. Transcriptome analysis indicated a group of up- and down-regulated genes in response to m6dA enrichment. Do all these genes harbor EcoDam and CcrM GATNTC recognition motif? If they do, what are the correlations between m6dA genomic locations (promoter, gene bodies, introns, exons) and gene expression?

4. From reporter assays, the authors suggested m6dA is a repressive marker that m6dA deposition drastically reduced their expression (Fig. 4). This is consistent with the notion that m6dA antagonizes JUN family TFs (Fig. 6). However, they also suggested m6dA can promote PRC2 regulated genes via reducing H3K37me3. How do the authors envision the differential roles of m6dA they found?

5. It will be helpful if they could provide JUN ChIP-seq, or at least qPCR data to validate JUN binding on the genes they identified? Whether depletion of JUN could directly affect these gene expression?

6. The authors should engage some discussion for the high vs low level of m6dA in regulating gene expressions. They can overexpress the N6-MTases to induce high m6dA, but this is rarely the case in the mammalian genome under physiological conditions.

Minor: Page 8, line 293, should be PRC2 not PCR2.

Reviewer #3 (Remarks to the Author):

An interesting and well written manuscript describing the effect of exogenously introduced m6dA on the biology of a transformed human cell line. The experimental design is simple, careful, robust and appropriate to the scientific question, but is highly artificial which questions the relevance of the results. I also found the paper confused in terms of its conclusions and motivation.

MAJOR COMMENTS

The authors begin by stating that 6mA in human DNA is controversial. The current state of the field suggests that if 6mA does get incorporated into human DNA it is indirectly via recycling of RNA. Moreover, regardless of mode of incorporation the amount of 6mA in DNA is exceptionally rare (as shown also here) and does not seem to occur in any given sequence context. Despite this the authors claim to be 'investigating the biological role of m6dA in human DNA' and claim that 'beyond doubt' that m6dA in human DNA has biological effects. I question the relevance of this:

1. Is it surprising that covalently binding a methyl group to adenines at a level 10,000 times greater than ever observed in normal tissues 'effects' biology? This does not mean that 6mdA has a biological role in mammals or that it even occurs in human DNA – Indeed, in the present study the authors cannot detect 6mdA in wild-type cells.

2. The authors use restriction digest to show that specific 'sequence contexts' are methylated, but this does not rule out the MTases being used are methylating elsewhere in the genome or that any given locus is methylated. This is particularly troublesome as the authors go on to make claims about m6dA affecting binding of transcription factors to specific sites in the genome without showing if these sites are in fact methylated. That these specific sites are methylated in the cells needs to be shown.

3. What is the more general biological relevance of these findings? If 6mA does get incorporated into human DNA at random at levels 10,000 times less than reported here, how often will that

incorporation occur at a specific TF binding site?

4. The RNA-Seq experiments are poorly described – how many replicates for example. The link to the data in the DaRUS database did not work. Also, why is the sequencing data not being deposited in a more well-known database (GEO, arrayexpress)?

5. Despite the known problems with contamination, the authors make no mention of how they ensured contamination did not affect their results, i.e. mycoplasma testing is absent.

6. The selection of reporter genes is vague – ‘multiple GANTC sites’. An exact definition of the unbiased selection needs to be provided.

7. A supplementary table with the list of the differentially expressed genes and their expression in each condition needs to be provided.

8. How the authors identified an ‘enrichment’ for JUN binding sites is not described in the methods and is critical.

9. In general I found the methods relating to bioinformatics analyses vague and unhelpful.

10. The conclusions are completely overdriven given that we are still unaware if 6mdA occurs in humans and need to be toned down significantly.

Reviewer #4 (Remarks to the Author):

Regarding that the controversy remained in the DNA m6dA field, it is a great challenge to elucidate physiological functions of DNA m6dA. To explore the potential roles and functions of m6dA in human cells, instead of human m6dA methylases, the authors introduce m6dA in human DNA using bacterial DNA methyltransferases. The transduced bacterial DNA methyltransferases generated Gm6dANTC and Gm6ATC sites at genome wide scale. By this stepwise approach, they showed: 1) GANTC/GATC methylation reduces cell viability; 2) 99 genes are found to be directly regulated by m6dA in a GANTC context; 3) Upregulated genes (33 genes) showed m6dA-dependent reduction of H3K27me3, probably suggesting that the PRC2 complex is inhibited by m6dA; 4) Genes downregulated by m6dA (66 genes) showed enrichment of JUN family transcription factor binding sites; and 5) JUN binds m6dA containing DNA with reduced affinity, suggesting that m6dA can reduce the recruitment of JUN transcription factors to target genes. Interestingly, they use a GFP reporter gene to show that that m6dA in some promoters can reduce gene expression, provide direct evidence on the regulation of the gene expression by m6dA. It is waiting to see in the future if truly happens in human cells with human methylases, which are yet unknown.

Major

1. What is the mechanism responsible for m6dA mediated proliferation inhibition?

2. Although they claimed “our study documents beyond doubt that m6dA in human DNA has biological effects”, they authors cannot show the presence of the orthologs of bacterial m6dA methyltransferases (EcoDam and CcrM) in mammals. Therefore, they cannot prove these observations are physiologically relevant. This is just a possibility. They should mitigate their claim.

3. Figure 3 shows CCRm-caused difference in gene expression. This difference is simply attributed to the Genome-wide deposition of m6dA by the bacterial CcrM MTase. However, they did not show the direct detection of m6dA in these DE genes. They should show experimental evidence on the presence of m6dA in these genes, especially the position.

4. It is puzzling that, although extremely high m6dA deposited on human DNA, the regulated genes are very few (99 genes). In bacteria, m6dA mainly guide gene replication and DNA mismatch repair. Is it possible that the deposited m6dA may affect DNA replication and repair rather than regulate gene

expression?

Minor

1. "this modification has been rediscovered in 2015 at low levels in several higher eukaryotes 4." Original Cell papers should be cited rather than the cited review article.
2. One latest paper regarding 6mdA misincorporation is missed (Lyu et al., Cell Discovery, 8:39 (2022)). This paper shows that misincorporated m6dA is a potential hallmark for human glioblastoma.

Response to the reviewers' comments

Reviewer #1

“The article from Broche et al. aims at describing functional relationships between gene expression and 6mdA deposition by bacterial DNA methyltransferases (EcoDam and CcrM) in H293 cells. The authors first successfully validate the expression of wild-type and mutant DNA methyltransferases as well as 6mdA deposition and identify a decrease of proliferation upon expression of these enzymes, linked to deposition of 6mdA. They further identify a very small set of genes (less than 100 genes) to be differentially expressed upon expression of wt CcrM. Amongst these, 7 of them are enriched with GANTC motifs (recognized by CcrM) and are expressed in a luciferase assay, of which 6 expressed at sufficient levels. The authors observed that 4 of those 6 genes are directly regulated by CcrM. Next, they aimed at finding mechanistic links between 6mdA deposition and gene expression, and found a handful of genes regulated by PRC2/H3K27me3 as well as with JUN family of transcription factors. Overall, I find the article to be of moderate interest, but acknowledge the need for understanding the dynamics of gene expression and 6mdA deposition in vivo.

My major concern is that H293 cells lack 6mdA (at least as identified by restriction assays and MS/MS in this manuscript). I initially thought that overexpressing bacterial DNA methyltransferases would help identify a subset of genes that would be regulated by 6mdA deposition, and that the authors would then quest for looking at endogenous 6mdA on these targets (for example by immunoprecipitation with an antibody raised against 6mA), but surprisingly it was not the case. Instead, they focused on examining regulation mechanisms on a very low number of targets. The purpose of studying such an artificial system is to link it to endogenous m6dA deposition, otherwise it seems to me that the authors are focusing on an artifact rather than true biology. Further, it is expected to have a handful of targets (99) differentially expressed upon deposition of an exogenous epigenetic mark, of which it is also not surprising that a few are PRC2 or JUN -regulated. In brief, unless Prof. Jeltsch's team can find natural m6dA deposited on a significant subset of the 99 deregulated targets, I unfortunately believe that the manuscript is not suitable for publication in Communications Biology.”

Reply: Thank you for working with our manuscript. As mentioned in the manuscript, the field of m6dA is very controversial, with critical discussion regarding key methods like generation of antibody based m6dA profiles, key findings (unclear levels of m6dA), and unclear pathways of potential m6dA incorporation into genomic DNA (by dedicated enzymes, through DNA replication or by both of these ways). It was not the aim of our work to contribute to these ongoing debates directly. Instead, it was the aim of our work to develop a cellular model system allowing to study potential effects of m6dA in human cells in a setting that clearly allows to attribute biological effects to the presence m6dA. For this, we introduced 6mdA artificially at GATC and GANTC motifs and showed a clear reduction of the viability of HEK293 cells after introduction of the m6dA. Hence our work clearly documents a potential biological effect of m6dA in human cells, which we believe is a very important finding. Please note, it also would have been possible to find absence of any effects as illustrated by the weaker effects observed with GATC methylation.

Starting from this initial observation, we then studied the molecular pathways leading to the reduced cell proliferation of HEK293 cells containing GANTC methylation. We show changes in gene expression connected to the JUN TF family and Polycomb silencing. Please note, that it is not correct to state that just a few genes in our list of differentially expressed genes were JUN or Polycomb targets, because we observed a strong and highly significant enrichment of these genes. Hence, we did identify groups of genes that are regulated by m6dA and identified molecular pathways mediating

this effect. Although our cellular system is an artificial experimental model, these are very relevant findings, as they exemplify molecular mechanisms of action m6dA can have in human cells.

Reviewer #2

“The presence and functions of DNA 6-methyladenine (m6dA) have been heavily debated in the recent years, and several studies concluded m6dA was purely artifact in the genome, due to potential bacteria contamination. However, others insisted the presence of m6dA in metazoans, despite in very low amount due to active or passive mechanisms, and their potential epigenetic functions. In this study, Broche et al took the biorthogonal approach to overexpress two well-established bacterial N6-MTases, EcoDam and CcrM, to monitor the roles of high m6dA deposition in human HEK293 cells genome. High level of m6dA, validated by m6dA sensitive restriction enzyme digestion and quantitative HPLC, significantly impair normal cell proliferation. Global m6dA alteration resulted in a group of significantly dysregulated genes that were potentially modulated by promoter m6dA deposition. M6dA could crosstalk with Polycomb/H3K27me3 and DNA binding JUN-family transcription factors to modulate gene expression in both endogenous loci and trans-genes. Overall, this is a well-tailored studies addressing a critical and confusing question – does m6dA indeed epigenetically regulate mammalian gene expression. One cannot ignore potential biological roles of m6dA simply because it is sparse in the genome. This study effectively contributes to our understanding of m6dA in mammalian genome, which could be transiently or loci-specifically enriched to a meaningful amount to exert epigenetic roles. However, several conceptual and technical gaps should be resolved before moving forward with this manuscript:”

Reply: Thank you for working with our manuscript and the insightful comments. We appreciate your conclusion that the manuscript is “a well-tailored studies addressing a critical and confusing question” and it “effectively contributes to our understanding of m6dA in mammalian genome”. We hope that your questions were convincingly answered and you can recommend acceptance of our revised manuscript.

“1. The authors designed N6-MTase expression constructs to separately express N6-MTase and GFP under DOX control, then mixed the GFP positive cells with parental 293 cells to monitor and compare their growth. However, GFP expression is under the control of IRES, which might not necessarily synchronize with the N6MTase expression. The enrichment of m6dA from GFP positive cells, comparing to no GFP cells, should be carefully tested. The co-expression of N6-MTase and GFP should be monitored by immunofluorescence.”

Reply: This point has been carefully examined. The m6dA levels in GFP negative HEK293 cells are reported in Figure 2B. These cells are identical to the GFP negative HEK293 cells shown in Fig. 1C and D. Data for GFP positive cells are given in Figure 2B as well as Suppl. Fig. 1 and 2. In Suppl. Fig. 1 and 2, we also compare methylation of the same cells without dox induction (=GFP negative) and after dox induction (=GFP positive), showing that the GFP signal and DNA methylation are fully controlled by dox. Moreover, we show after 10 days of dox treatment no reduction of the genome m6dA levels, indicting the absence of strong counterselection against MTase expression. Hence, we clearly document at the population level a co-expression of MTase (by DNA methylation analysis) and GFP (by cell sorting) as requested by the reviewer. Finally, we like to mention that a potential decoupling of GFP and MTase expression would always reduce the size of our effects, hence the observed effects are fully reliable.

“2. While EcoDam and CcrM expression both drive the m6dA production, it is plausible they regulate different subset of m6dA deposition to specific loci. Particularly, CcrM displayed much more strong cellular effects in HEK293 cell proliferation and was likely to largely responsible for the synergistic effects of EcoDam and CcrM co-expression (Fig. 2C). It will be much more informative if the authors could provide genome-wide m6dA profile, or at least validate m6dA on several key loci to delineate loci-specific roles of these methyltransferases.”

Reply: We perfectly agree with your statement that CcrM dominates the combined effects of CcrM and EcoDam. Please note that this was also concluded in our manuscript at the end of the chapter “Combined expression of EcoDam and CcrM in HEK293 cells leads to increased m6dA levels” where it was stated: “Overall, these data clearly indicate that the gain of methylation at GANTC sequences has a substantially higher impact on cell proliferation than methylation of GATC sequences...” This was also our reason to focus our study of the biological effect on CcrM methylation.

Regarding the m6dA profile, we understand your point and this question was on our initial project plans as well. However, our LC-MS/MS mass spec analyses (Fig. 2B) as well as the restriction digestion studies (Suppl. Fig. 1 and 2) clearly show an almost quantitative methylation of the genomic DNA at GATC and GANTC motifs. Based on this, a further analysis of the genome-wide m6dA profile does not make sense, because it would only reproduce the distribution of the corresponding motifs in the genome. The important finding that GATC and GANTC motifs were completely methylated in a genome-wide fashion has been further clarified in the revised version of the manuscript.

“3. Transcriptome analysis indicated a group of up- and down-regulated genes in response to m6dA enrichment. Do all these genes harbor EcoDam and CcrM GATNTC recognition motif? If they do, what are the correlations between m6dA genomic locations (promoter, gene bodies, introns, exons) and gene expression?”

Reply: We could not find any meaningful enrichment or depletion of GANTC sites. However, please note that GANTC sites in the human genome occur every 256 base pairs on average. Promoters are usually considered to be several kb long, even not considering associated enhancers, and Polycomb silencing can be triggered at kb distances from genes as well. Hence a more detailed analysis of the enrichment of these sites in these functional regions is not promising. We have now indicated in the manuscript that all differentially regulated genes contain at least one GANTC site in the promoter region and described the distribution (average 7.1 ± 3.2 SD).

“4. From reporter assays, the authors suggested m6dA is a repressive marker that m6dA deposition drastically reduced their expression (Fig. 4). This is consistent with the notion that m6dA antagonizes JUN family TFs (Fig. 6). However, they also suggested m6dA can promote PRC2 regulated genes via reducing H3K37me3. How do the authors envision the differential roles of m6dA they found?”

Reply: Thank you for this hint. Downregulation of genes can be explained by the inhibition of DNA binding of activating factors (JUN). Upregulation of genes can be explained by the inhibition of the activity of repressive factors (PRC2). We have added a sentence in the discussion to clarify this point.

“5. It will be helpful if they could provide JUN ChIP-seq, or at least qPCR data to validate JUN binding on the genes they identified? Whether depletion of JUN could directly affect these gene expression?”

Reply: We understand your question. However, please note, that the JUN/FOS group of TFs is quite large and all members bind to TGANTCA motifs, examples of this are shown in Fig. 6A. As we did not know the relevant factor, it was not possible to conduct targeted ChIP analyses.

However, we showed in Fig. 6C and D that expression of the EMILIN2 promoter (in a reporter gene setting) was indeed dependent on an intact JUN binding motif, while the repression of expression by m6dA was dependent on an intact GANTC motif. These data document the role of JUN and GANTC motifs in the m6dA mediated silencing processes observed in our study for this example promoter. This point has been explained more explicitly in the manuscript.

“6. The authors should engage some discussion for the high vs low level of m6dA in regulating gene expressions. They can overexpress the N6-MTases to induce high m6dA, but this is rarely the case in the mammalian genome under physiological conditions.”

Reply: Thank you for this hint. We have added one sentence to clarify that the gene regulatory mechanisms discovered here only require the local methylation of few GANTC motifs. Hence, they could operate, even if global m6dA levels are low.

“Minor: Page 8, line 293, should be PRC2 not PCR2.”

Reply: Corrected. Thank you.

Reviewer #3

“An interesting and well written manuscript describing the effect of exogenously introduced m6dA on the biology of a transformed human cell line. The experimental design is simple, careful, robust and appropriate to the scientific question, but is highly artificial which questions the relevance of the results. I also found the paper confused in terms of its conclusions and motivation.”

Reply: Thank you for working with our manuscript and the insightful comments. We appreciate your statement that the paper is “interesting and well written”.

“MAJOR COMMENTS

The authors begin by stating that 6mA in human DNA is controversial. The current state of the field suggests that if 6mA does get incorporated into human DNA it is indirectly via recycling of RNA. Moreover, regardless of mode of incorporation the amount of 6mA in DNA is exceptionally rare (as shown also here) and does not seem to occur in any given sequence context. Despite this the authors claim to be ‘investigating the biological role of m6dA in human DNA’ and claim that ‘beyond doubt’ that m6da in human DNA has biological effects. I question the relevance of this:”

Reply: Thank you for these helpful comments. We have modified the writing towards “potential biological role”. Moreover, we changed wording towards “can have”. These (or similar) changes were implemented at several places

“1. Is it surprising that covalently binding a methyl group to adenines at a level 10,000 times greater than ever observed in normal tissues ‘effects’ biology? This does not mean that 6mdA has a

biological role in mammals or that it even occurs in human DNA – Indeed, in the present study the authors cannot detect 6mdA in wild-type cells.”

Reply: As mentioned in the manuscript, the field of m6dA is very controversial with high variation of reported results between cell lines/tissues, development and genomic regions. Without going into details and knowing that these numbers are also controversial due to potential contaminations, we like to mention that levels up to 1000 ppm were reported in specific chromatin regions (Xiao et al., 2018, *Molecular cell* 71 (2):306-318 e307). We observed global levels of about 10,000 ppm. Hence, the discrepancy is not that high, if we compare the local enrichment. Having the controversial interpretation of the m6dA measurements in mind, we prefer not to add this estimation to the manuscript.

It was the aim of our work to develop a cellular model system allowing to study potential effects of m6dA in human cells directly. Using HEK293 cells with very low endogenous m6dA levels may even be beneficial for our experiments as the data will only represent the exogenously introduced methylation. As next step, we introduced 6mdA artificially at GATC and GANTC motifs and showed a clear reduction of the viability of the HEK293 cells after incorporation of the m6dA. With this approach we clearly document a biological effect of m6dA in human cells, which we believe is a very important finding. Please note, it also would have been possible to find absence of any effects, as illustrated by the weaker effects observed with GATC methylation. Starting from this point, we then studied the molecular pathways leading to the reduced cell proliferation of HEK293 cells containing GANTC methylation. We show changes in gene expression connected to the JUN TF family and Polycomb silencing. Although our cellular system is an artificial experimental model, these are very relevant findings, as they exemplify potential molecular mechanisms of action of m6dA in human cells.

Additionally, we like to mention that the gene regulatory mechanisms discovered here only require the local methylation of few GANTC motifs. Hence, they could operate, even if global m6dA levels are low. This point has now been mentioned in the manuscript.

“2. The authors use restriction digest to show that specific ‘sequence contexts’ are methylated, but this does not rule out the MTases being used are methylating elsewhere in the genome or that any given locus is methylated. This is particularly troublesome as the authors go on to make claims about m6dA affecting binding of transcription factors to specific sites in the genome without showing if these sites are in fact methylated. That these specific sites are methylated in the cells needs to be shown.”

Reply: Please note that the restriction digestion analyses (Suppl. Fig. 1 and 2) do show the full genome-wide methylation of GANTC and GATC motifs after the expression of the corresponding MTases. This conclusion is further validated by the mass spectrometry data (Fig. 2B). This implies that the GANTC motifs associated with differentially methylated genes are methylated as well, hence your requirement is fulfilled. Given the genome-wide deposition of m6dA in the target motifs, further m6dA mapping would be pointless and just reproduce the occurrence of GANTC motifs.

Regarding OFF-target methylation of the MTases: The specificities of both bacterial enzymes have been extensively studied and they methylate DNA with very high preferences for their target sites with few non-cognate sites with 100-1000 fold reduced methylation, most other sites showing >10,000 fold reduced methylation (EcoDam doi:10.1016/j.jmb.2006.02.028; CcrM doi:10.1093/nar/gkr768). The high specificity of the genomic DNA-adenine-methylation has now also

been validated experimentally by control restriction digestions showing the absence of methylation in OFF-target motifs which are now presented in Suppl. Fig. 1.

“3. What is the more general biological relevance of these findings? If 6mA does get incorporated into human DNA at random at levels 10,000 times less than reported here, how often will that incorporation occur at a specific TF binding site?”

Reply: Thank you for this hint. As mentioned already, we have added a sentence in the discussion that takes up this point: “In the biological context, our data indicate that targeted introduction of m6dA at GANTC sites in PRC2 or JUN binding sites could directly up- or down-regulate the expression of associated genes, even if the overall global levels of m6dA were low.” Of course, “random” deposition of m6dA would just lead to biological noise.

4. The RNA-Seq experiments are poorly described – how many replicates for example. The link to the data in the DaRUS database did not work. Also, why is the sequencing data not being deposited in a more well-known database (GEO, arrayexpress)?

Reply: We like to apologize for the missing information. The RNA-seq experiments were done in duplicates for each condition. This information has been added to the text. In addition, method descriptions were expanded. The DARUS link was tested by us several times, perhaps the problem was related to the line break. We have now reformatted the paragraph to provide the link without line break and it also included here:

<https://darus.uni-stuttgart.de/privateurl.xhtml?token=408a970a-ea5f-43f4-b4ac-bf318c4d9e6b>

The final stable DOI link to the entry will be <https://doi.org/10.18419/darus-2813> which is short and more convenient.

5. Despite the known problems with contamination, the authors make no mention of how they ensured contamination did not affect their results, i.e. mycoplasma testing is absent.

Reply: Due to our controls, it is beyond doubt that the high levels of cellular m6dA were introduced by the overexpressed MTases in our experiments. The mycoplasma testing was specified in the NR reporting document. The corresponding information has now been included into the method section of the manuscript as well.

6. The selection of reporter genes is vague – ‘multiple GANTC sites’. An exact definition of the unbiased selection needs to be provided.

Reply: The corresponding chapter has been rewritten and clarified. All promoters contained several GANTC sites which now has been mentioned. The selection of example promoters was based on the RNA-seq data.

7. A supplementary table with the list of the differentially expressed genes and their expression in each condition needs to be provided.

Reply: This has been done.

8. How the authors identified an 'enrichment' for JUN binding sites is not described in the methods and is critical.

Reply: Thank you for this comment and (again) we apologize for a mistake. The analysis of up- and downregulated genes was done with Enrichr. Unfortunately, during manuscript rewriting, both chapters were separated and the description and reference pointing to Enrichr and Fig. 5A was lost in the section about the discovery of the enrichment of JUN binding sites. This has now been corrected.

9. In general I found the methods relating to bioinformatics analyses vague and unhelpful.

Reply: We hope that the changes introduced in response to your comments #4, 6 and 8 could clarify many points.

10. The conclusions are completely overdriven given that we are still unaware if 6mdA occurs in humans and need to be toned down significantly.

Reply: Thank you for this hint. We have now mentioned at several places that the m6dA studied here was introduced "artificially". Moreover, the last sentence of the discussion now refers to "potential m6dA readers and human N6-MTases". Additional changes have already been described above before point #1.

Reviewer #4

Regarding that the controversy remained in the DNA m6dA field, it is a great challenge to elucidate physiological functions of DNA m6dA. To explore the potential roles and functions of m6dA in human cells, instead of human m6dA methylases, the authors introduce m6dA in human DNA using bacterial DNA methyltransferases. The transduced bacterial DNA methyltransferases generated Gm6dANTC and Gm6ATC sites at genome wide scale. By this stepwise approach, they showed: 1) GANTC/GATC methylation reduces cell viability; 2) 99 genes are found to be directly regulated by m6dA in a GANTC context; 3) Upregulated genes (33 genes) showed m6dA-dependent reduction of H3K27me3, probably suggesting that the PRC2 complex is inhibited by m6dA; 4) Genes downregulated by m6dA (66 genes) showed enrichment of JUN family transcription factor binding sites; and 5) JUN binds m6dA containing DNA with reduced affinity, suggesting that m6dA can reduce the recruitment of JUN transcription factors to target genes.

Interestingly, they use a GFP reporter gene to show that that m6dA in some promoters can reduce gene expression, provide direct evidence on the regulation of the gene expression by m6dA. It is waiting to see in the future if truly happens in human cells with human methylases, which are yet unknown.

Reply: Thank you for working with our manuscript and the insightful comments.

Major

1. What is the mechanism responsible for m6dA mediated proliferation inhibition?

Reply: We have identified the altered regulation of at least 99 genes and molecular pathways of m6dA leading to up- and downregulation of genes. Finding out which of the affected genes is responsible for the change in HEK293 viability in our experimental settings would require additional experiments that are not straightforward in particular as the effects could also arise from combinations of genes. Hence this work would be beyond the scope of our current manuscript.

2. Although they claimed “our study documents beyond doubt that m6dA in human DNA has biological effects”, they authors cannot show the presence of the orthologs of bacterial m6dA methyltransferases (EcoDam and CcrM) in mammals. Therefore, they cannot prove these observations are physiologically relevant. This is just a possibility. They should mitigate their claim.

Reply: Thank you for this hint. We have changed writing at several places now indicating that the methylation was introduced artificially and we only identify potential functions m6dA could have in human cells. The sentences cited above has been changed to state it “can have biological effects”.

3. Figure 3 shows CcrM-caused difference in gene expression. This difference is simply attributed to the Genome-wide deposition of m6dA by the bacterial CcrM MTase. However, they did not show the direct detection of m6dA in these DE genes. They should show experimental evidence on the presence of m6dA in these genes, especially the position.

Reply: All DE genes contain GANTC sites in their promoters. This point has now been mentioned in the manuscript and we also specify the distribution of sites (average 7.1 ± 3.2 SD). The restriction digestion analyses (Suppl. Fig. 1 and 2) document a full genome-wide methylation of GANTC and GATC motifs after the expression of the corresponding MTases. This conclusion is further validated by the mass spectrometry data (Fig. 2B). This point has been clarified in the manuscript. This result implies that the GANTC motifs associated with DE genes are methylated, hence your requirement is fulfilled. Given the genome-wide deposition of m6dA in the target motifs, further m6dA mapping would be pointless as it would only reproduce the occurrence patterns of GANTC motifs.

“4. It is puzzling that, although extremely high m6dA deposited on human DNA, the regulated genes are very few (99 genes). In bacteria, m6dA mainly guide gene replication and DNA mismatch repair. Is it possible that the deposited m6dA may affect DNA replication and repair rather than regulate gene expression?”

Reply: Thank you. We had similar thoughts, but the profile of the misregulated genes gives no hints towards DNA replication or DNA repair. As we believe problems in one of these fundamental processes would lead to characteristic secondary responses in gene expression patterns, we cannot propose that these processes are altered on the basis of our data.

“Minor

1. “this modification has been rediscovered in 2015 at low levels in several higher eukaryotes 4.”
Original Cell papers should be cited rather than the cited review article.”

Reply: These references are now given as well. Due to the very controversial situation, we believe providing some review that summarize the state of the field if of relevance for readers as well.

“2. One latest paper regarding 6mdA misincorporation is missed (Lyu et al., Cell Discovery, 8:39 (2022)). This paper shows that misincorporated m6dA is a potential hallmark for human glioblastoma.”

Reply: Thank you. This work has now been cited as well. We also added two more citations, which escaped our attention in the first round of writing.

Li, Z., et al., Nature, 2020. 583(7817): p. 625-630.

Fernandes, S.B., et al., Front Genet, 2021. 12: p. 657171.

Response to the reviewers' comments

Comment for all reviewers

Thank you for working with our manuscript. Following your valuable comments, we modified the text at several places. Moreover, we added new data to document the motif specificity of the introduced CcrM and EcoDam methylation (new panel in Suppl. Fig. 1) and we experimentally validated the presence of m6dA in genomic DNA isolated from cells after CcrM expression at the four m6dA sensitive promoters identified in our work (EMILIN2, ECM1, FAM46B, EGF) (new Suppl. Fig. 6).

Reviewer #1

“The article from Broche et al. aims at describing functional relationships between gene expression and 6mdA deposition by bacterial DNA methyltransferases (EcoDam and CcrM) in H293 cells. The authors first successfully validate the expression of wild-type and mutant DNA methyltransferases as well as 6mdA deposition and identify a decrease of proliferation upon expression of these enzymes, linked to deposition of 6mdA. They further identify a very small set of genes (less than 100 genes) to be differentially expressed upon expression of wt CcrM. Amongst these, 7 of them are enriched with GANTC motifs (recognized by CcrM) and are expressed in a luciferase assay, of which 6 expressed at sufficient levels. The authors observed that 4 of those 6 genes are directly regulated by CcrM. Next, they aimed at finding mechanistic links between 6mdA deposition and gene expression, and found a handful of genes regulated by PCR2/H3K27me3 as well as with JUN family of transcription factors. Overall, I find the article to be of moderate interest, but acknowledge the need for understanding the dynamics of gene expression and 6mdA deposition in vivo.

My major concern is that H293 cells lack 6mdA (at least as identified by restriction assays and MS/MS in this manuscript). I initially thought that overexpressing bacterial DNA methyltransferases would help identify a subset of genes that would be regulated by 6mdA deposition, and that the authors would then quest for looking at endogenous 6mdA on these targets (for example by immunoprecipitation with an antibody raised against 6mA), but surprisingly it was not the case. Instead, they focused on examining regulation mechanisms on a very low number of targets. The purpose of studying such an artificial system is to link it to endogenous m6dA deposition, otherwise it seems to me that the authors are focusing on an artifact rather than true biology. Further, it is expected to have a handful of targets (99) differentially expressed upon deposition of an exogenous epigenetic mark, of which it is also not surprising that a few are PRC2 or JUN -regulated. In brief, unless Prof. Jeltsch's team can find natural m6dA deposited on a significant subset of the 99 deregulated targets, I unfortunately believe that the manuscript is not suitable for publication in Communications Biology.”

Reply: Thank you for working with our manuscript. As mentioned in the manuscript, the field of m6dA is very controversial, with critical discussions regarding key methods (like generation of antibody based m6dA profiles), key findings (unclear levels of m6dA), and unclear pathways of potential m6dA incorporation into genomic DNA (by dedicated enzymes, through DNA replication or by both of these ways). It was not the aim of our work to contribute to these ongoing debates directly. Instead, we intended to develop a cellular model system allowing to study potential effects of m6dA in human cells in a setting that clearly allows to attribute biological effects to the presence m6dA. For this, we introduced m6dA artificially at GATC and GANTC motifs and showed a concomitant reduction of the viability of HEK293 cells. Hence our work clearly documents a direct biological effect of m6dA in human cells, which we believe is a very important finding. Please note, it also would have been possible to observe an absence of any effects.

Starting from this initial observation, we then studied the molecular pathways leading to the reduced cell proliferation of HEK293 cells containing GANTC methylation. We show changes in gene expression connected to the JUN transcription factor binding and Polycomb silencing. Please note, that it is not correct to state that just a few genes in our list of differentially expressed genes were JUN or Polycomb targets, because we observed a strong and highly significant enrichment of these genes among the down- or upregulated genes. Hence, we did identify groups of genes that are regulated by m6dA in human cells and identified molecular pathways mediating this effect. Although our cellular system is an artificial experimental model, these are very relevant findings, as they exemplify molecular mechanisms of action endogenous m6dA can have in human cells.

Reviewer #2

“The presence and functions of DNA 6-methyladenine (m6dA) have been heavily debated in the recent years, and several studies concluded m6dA was purely artifact in the genome, due to potential bacteria contamination. However, others insisted the presence of m6dA in metazoans, despite in very low amount due to active or passive mechanisms, and their potential epigenetic functions. In this study, Broche et al took the biorthogonal approach to overexpress two well-established bacterial N6-MTases, EcoDam and CcrM, to monitor the roles of high m6dA deposition in human HEK293 cells genome. High level of m6dA, validated by m6dA sensitive restriction enzyme digestion and quantitative HPLC, significantly impair normal cell proliferation. Global m6dA alteration resulted in a group of significantly dysregulated genes that were potentially modulated by promoter m6dA deposition. M6dA could crosstalk with Polycomb/H3K27me3 and DNA binding JUN-family transcription factors to modulate gene expression in both endogenous loci and trans-genes. Overall, this is a well-tailored studies addressing a critical and confusing question – does m6dA indeed epigenetically regulate mammalian gene expression. One cannot ignore potential biological roles of m6dA simply because it is sparse in the genome. This study effectively contributes to our understanding of m6dA in mammalian genome, which could be transiently or loci-specifically enriched to a meaningful amount to exert epigenetic roles. However, several conceptual and technical gaps should be resolved before moving forward with this manuscript:”

Reply: Thank you for working with our manuscript and the insightful comments. We appreciate your conclusion that the manuscript is “a well-tailored studies addressing a critical and confusing question” and it “effectively contributes to our understanding of m6dA in mammalian genome”. We hope that your questions were convincingly answered by the revision and you can recommend acceptance of our revised manuscript.

“1. The authors designed N6-MTase expression constructs to separately express N6-MTase and GFP under DOX control, then mixed the GFP positive cells with parental 293 cells to monitor and compare their growth. However, GFP expression is under the control of IRES, which might not necessarily synchronize with the N6MTase expression. The enrichment of m6dA from GFP positive cells, comparing to no GFP cells, should be carefully tested. The co-expression of N6-MTase and GFP should be monitored by immunofluorescence.”

Reply: This point has been carefully examined. The m6dA levels in GFP negative HEK293 cells are reported in Figure 2B. These cells are identical to the GFP negative HEK293 cells shown in Fig. 1C and D. Data for GFP positive cells are given in Figure 2B as well as Suppl. Fig. 1 and 2. In Suppl. Fig. 1 and 2, we also compare methylation of the same cells without dox induction (=GFP negative) and after dox induction (=GFP positive), showing that the GFP signal and DNA methylation are fully controlled

by dox. Moreover, we show after 10 days of dox treatment no reduction of the genomic m6dA levels, indicting the absence of strong counterselection against MTase expression. Hence, we clearly document the co-expression of MTase (by DNA methylation analysis) and GFP (by cell sorting) as requested by the reviewer. Finally, we like to mention that a hypothetical decoupling of GFP and MTase expression would always reduce the size of our effects, because MTase expression causes the biological effects while GFP is just the reporter. Hence, the observed effects are fully reliable.

“2. While EcoDam and CcrM expression both drive the m6dA production, it is plausible they regulate different subset of m6dA deposition to specific loci. Particularly, CcrM displayed much more strong cellular effects in HEK293 cell proliferation and was likely to largely responsible for the synergistic effects of EcoDam and CcrM co-expression (Fig. 2C). It will be much more informative if the authors could provide genome-wide m6dA profile, or at least validate m6dA on several key loci to delineate loci-specific roles of these methyltransferases.”

Reply: We perfectly agree with your statement that CcrM dominates the combined effects of CcrM and EcoDam. Please note that this was also concluded in our manuscript at the end of the chapter “Combined expression of EcoDam and CcrM in HEK293 cells leads to increased m6dA levels” where it was stated: “Overall, these data clearly indicate that the gain of methylation at GANTC sequences has a substantially higher impact on cell proliferation than methylation of GATC sequences...” This was also our reason to focus the next part of our study investigating the biological effects of m6dA on CcrM methylation.

Regarding the m6dA profile, we understand your point and this question was on our initial project plans as well. However, our LC-MS/MS analyses (Fig. 2B) as well as the restriction digestion studies (Suppl. Fig. 1 and 2) clearly show an almost quantitative methylation of the genomic DNA at GATC and GANTC motifs after expression of EcoDam or CcrM WT. Based on this, a further analysis of the genome-wide m6dA profile does not make sense, because it would only reproduce the distribution of the corresponding motifs in the genome. The important finding that GATC and GANTC motifs were completely methylated in a genome-wide fashion has been further clarified in the revised version of the manuscript.

To further validate the global GANTC methylation, we now also tested the presence of m6dA in newly added restriction protection-qPCR data showing full methylation of genomic DNA after expression of CcrM at the identified m6dA sensitive promoters (EMILIN2, ECM1, FAM46B, EGF). These data are now shown in the new Suppl. Fig. 6.

“3. Transcriptome analysis indicated a group of up- and down-regulated genes in response to m6dA enrichment. Do all these genes harbor EcoDam and CcrM GATNTC recognition motif? If they do, what are the correlations between m6dA genomic locations (promoter, gene bodies, introns, exons) and gene expression?”

Reply: We could not find any meaningful enrichment or depletion of GANTC sites. Statistically, GANTC (and GATC) sites occur in the human genome every 256 base pairs on average. Promoters are usually considered to be several kb long, even not taking into account the associated enhancers. Similarly, Polycomb silencing can be triggered at kb distances from genes as well. Hence, a more detailed analysis of the enrichment of GANTC sites in these functional regions is not promising. We have now indicated in the manuscript that all differentially regulated genes contain at least one GANTC site in the promoter region and mention the distribution (average 7.1 ± 3.2 SD).

“4. From reporter assays, the authors suggested m6dA is a repressive marker that m6dA deposition drastically reduced their expression (Fig. 4). This is consistent with the notion that m6dA antagonizes JUN family TFs (Fig. 6). However, they also suggested m6dA can promote PRC2 regulated genes via reducing H3K37me3. How do the authors envision the differential roles of m6dA they found?”

Reply: Thank you for this hint. Downregulation of genes can be explained by the inhibition of DNA binding of activating factors (JUN). Upregulation of genes can be explained by the inhibition of the activity of repressive factors (PRC2). We have added a sentence in the discussion to clarify this point.

“5. It will be helpful if they could provide JUN ChIP-seq, or at least qPCR data to validate JUN binding on the genes they identified? Whether depletion of JUN could directly affect these gene expression?”

Reply: We understand your question. However, please note, that the JUN/FOS group of TFs is quite large and all members bind to TGANTCA motifs, examples of this are shown in Fig. 6A. As we did not know the relevant factor, it was not possible to conduct targeted ChIP analyses.

However, we showed in Fig. 6C and D that expression of the EMILIN2 promoter (in a reporter gene setting) was indeed dependent on an intact JUN binding motif, while the repression of expression by m6dA was dependent on an intact GANTC motif. These data document the role of JUN and GANTC motifs in the m6dA mediated silencing processes observed in our study for this example promoter. This point has been explained more explicitly in the manuscript.

“6. The authors should engage some discussion for the high vs low level of m6dA in regulating gene expressions. They can overexpress the N6-MTases to induce high m6dA, but this is rarely the case in the mammalian genome under physiological conditions.”

Reply: Thank you for this hint. We have added one sentence to clarify that the gene regulatory mechanisms discovered here only require the local methylation of few GANTC motifs. Hence, they could operate, even if global m6dA levels are low.

“Minor: Page 8, line 293, should be PRC2 not PCR2.”

Reply: Corrected. Thank you.

Reviewer #3

“An interesting and well written manuscript describing the effect of exogenously introduced m6dA on the biology of a transformed human cell line. The experimental design is simple, careful, robust and appropriate to the scientific question, but is highly artificial which questions the relevance of the results. I also found the paper confused in terms of its conclusions and motivation.”

Reply: Thank you for working with our manuscript and the insightful comments. We appreciate your statement that the paper is “interesting and well written”.

“MAJOR COMMENTS

The authors begin by stating that 6mA in human DNA is controversial. The current state of the field suggests that if 6mA does get incorporated into human DNA it is indirectly via recycling of RNA. Moreover, regardless of mode of incorporation the amount of 6mA in DNA is exceptionally rare (as shown also here) and does not seem to occur in any given sequence context. Despite this the authors claim to be ‘investigating the biological role of m6dA in human DNA’ and claim that ‘beyond doubt’ that m6da in human DNA has biological effects. I question the relevance of this:”

Reply: Thank you for these helpful comments. We have modified the writing towards “potential biological role”. Moreover, we changed wording towards “can have”. These (or similar) changes were implemented at several places

“1. Is it surprising that covalently binding a methyl group to adenines at a level 10,000 times greater than ever observed in normal tissues ‘effects’ biology? This does not mean that 6mdA has a biological role in mammals or that it even occurs in human DNA – Indeed, in the present study the authors cannot detect 6mdA in wild-type cells.”

Reply: As mentioned in the manuscript, the field of m6dA is very controversial with high variation of reported results between cell lines/tissues, development and genomic regions. Without going into details and knowing that these numbers are also controversial due to potential contaminations, we like to mention that levels up to 1000 ppm were reported in specific chromatin regions (Xiao et al., 2018, Molecular cell 71 (2):306-318 e307). We observed global levels of about 10,000 ppm. Hence, the discrepancy is not that high, if we compare the local enrichment.

It was the aim of our work to develop a cellular model system allowing to study potential effects of m6dA in human cells directly. Using HEK293 cells with very low endogenous m6dA levels was beneficial for our experiments as the data will only represent the exogenously introduced methylation. As next step, we introduced 6mdA artificially at GATC and GANTC motifs and showed a reduction of the viability of the HEK293 cells after incorporation of the m6dA. With this approach we clearly document a biological effect of m6dA in human cells, which we believe is a very important finding. Please note, it also would have been possible to observe an absence of any effects. Starting from this point, we then studied the molecular pathways leading to the reduced cell proliferation of HEK293 cells containing GANTC methylation. We show changes in gene expression connected to the JUN transcription factor family and Polycomb silencing. Although our cellular system is an artificial experimental model, these are very relevant findings, as they exemplify potential molecular mechanisms of action endogenous m6dA could have in human cells.

Additionally, we like to mention that the gene regulatory mechanisms discovered here in principle only require the local methylation of few GANTC motifs. Hence, they could operate, even if global m6dA levels are low. This point has now been mentioned in the manuscript.

“2. The authors use restriction digest to show that specific ‘sequence contexts’ are methylated, but this does not rule out the MTases being used are methylating elsewhere in the genome or that any given locus is methylated. This is particularly troublesome as the authors go on to make claims about m6dA affecting binding of transcription factors to specific sites in the genome without showing if these sites are in fact methylated. That these specific sites are methylated in the cells needs to be shown.”

Reply: Please note that the restriction digestion analyses (Suppl. Fig. 1 and 2) do show the full genome-wide methylation of GANTC and GATC motifs after the expression of the corresponding MTases. This conclusion is further validated by the mass spectrometry data (Fig. 2B). This implies that the GANTC motifs associated with differentially methylated genes are methylated as well. This conclusion was further validated in newly added qPCR data showing full GANTC methylation in the genomic DNA after CcrM expression at the identified m6dA sensitive promoters (EMILIN2, ECM1, FAM46B, EGF). These data are now presented in the new Suppl. Fig. 6. Given the genome-wide deposition of m6dA in the target motifs, further m6dA mapping would be pointless and just reproduce the occurrence of GANTC motifs.

Regarding OFF-target methylation of the MTases: The specificities of both bacterial enzymes have been extensively studied and they methylate DNA with very high preferences for their target sites with few non-cognate sites with 100-1000 fold reduced methylation, most other sites showing >10,000 fold reduced methylation (EcoDam doi:10.1016/j.jmb.2006.02.028; CcrM doi:10.1093/nar/gkr768). The high specificity of the genomic DNA-adenine-methylation has now also been validated experimentally by control restriction digestions showing the absence of methylation in OFF-target motifs which are now included in Suppl. Fig. 1.

“3. What is the more general biological relevance of these findings? If 6mA does get incorporated into human DNA at random at levels 10,000 times less than reported here, how often will that incorporation occur at a specific TF binding site?”

Reply: Thank you for this hint. We have added a sentence in the discussion that takes up this point: “In the biological context, our data indicate that targeted introduction of m6dA at GANTC sites in PRC2 or JUN binding sites could directly up- or down-regulate the expression of associated genes, even if the overall global levels of m6dA were low.” Of course, “random” deposition of m6dA would just lead to biological noise.

4. The RNA-Seq experiments are poorly described – how many replicates for example. The link to the data in the DaRUS database did not work. Also, why is the sequencing data not being deposited in a more well-known database (GEO, arrayexpress)?

Reply: We like to apologize for the missing information. The RNA-seq experiments were done in duplicates for each condition. This information has been added to the text. In addition, method descriptions were expanded. The DARUS link was tested by us several times, perhaps the problem was related to the line break. We have now reformatted the paragraph to provide the link without line break and it has been included here as well:

<https://darus.uni-stuttgart.de/privateurl.xhtml?token=408a970a-ea5f-43f4-b4ac-bf318c4d9e6b>

The final stable DOI link to the entry will be <https://doi.org/10.18419/darus-2813> which is short and more convenient.

5. Despite the known problems with contamination, the authors make no mention of how they ensured contamination did not affect their results, i.e. mycoplasma testing is absent.

Reply: Due to our controls with “No dox” and “inactive MTases”, it is beyond doubt that the high levels of cellular m6dA were introduced by the overexpressed MTases in our experiments. The

mycoplasma testing was specified in the NR reporting document. The corresponding information has now been included into the method section of the manuscript as well.

6. The selection of reporter genes is vague – ‘multiple GANTC sites’. An exact definition of the unbiased selection needs to be provided.

Reply: The corresponding chapter has been rewritten and clarified. All promoters contained several GANTC sites which now has been mentioned. The selection of example promoters was based on the RNA-seq data.

7. A supplementary table with the list of the differentially expressed genes and their expression in each condition needs to be provided.

Reply: This has been done.

8. How the authors identified an ‘enrichment’ for JUN binding sites is not described in the methods and is critical.

Reply: Thank you for this comment and (again) we apologize for the mistake. The analysis of up- and downregulated genes was done with Enrichr. Unfortunately, during manuscript rewriting, both chapters were separated and the descriptions and reference pointing to Enrichr and Fig. 5A were lost in the section about the discovery of the enrichment of JUN binding sites. This has now been corrected.

9. In general I found the methods relating to bioinformatics analyses vague and unhelpful.

Reply: We hope that the changes introduced in response to your comments #4, 6 and 8 could clarify many points.

10. The conclusions are completely overdriven given that we are still unaware if m6dA occurs in humans and need to be toned down significantly.

Reply: Thank you for this hint. We have now mentioned at several places that the m6dA studied here was introduced “artificially”. Moreover, the last sentence of the discussion now refers to “potential m6dA readers and human N6-MTases”. Additional changes have already been described above before point #1.

Reviewer #4

Regarding that the controversy remained in the DNA m6dA field, it is a great challenge to elucidate physiological functions of DNA m6dA. To explore the potential roles and functions of m6dA in human cells, instead of human m6dA methylases, the authors introduce m6dA in human DNA using bacterial DNA methyltransferases. The transduced bacterial DNA methyltransferases generated Gm6dANTC and Gm6ATC sites at genome wide scale. By this stepwise approach, they showed: 1) GANTC/GATC methylation reduces cell viability; 2) 99 genes are found to be directly regulated by m6dA in a GANTC

context; 3) Upregulated genes (33 genes) showed m6dA-dependent reduction of H3K27me3, probably suggesting that the PRC2 complex is inhibited by m6dA; 4) Genes downregulated by m6dA (66 genes) showed enrichment of JUN family transcription factor binding sites; and 5) JUN binds m6dA containing DNA with reduced affinity, suggesting that m6dA can reduce the recruitment of JUN transcription factors to target genes.

Interestingly, they use a GFP reporter gene to show that that m6dA in some promoters can reduce gene expression, provide direct evidence on the regulation of the gene expression by m6dA. It is waiting to see in the future if truly happens in human cells with human methylases, which are yet unknown.

Reply: Thank you for working with our manuscript and the insightful comments.

Major

1. What is the mechanism responsible for m6dA mediated proliferation inhibition?

Reply: We have identified the altered regulation of at least 99 genes and molecular pathways of m6dA leading to up- and downregulation of genes. Finding out which of the affected genes is responsible for the change in HEK293 viability in our experimental settings would require additional experiments that are not straightforward in particular as the effects could also arise from combinations of genes. Hence this work would be beyond the scope of our current manuscript.

2. Although they claimed “our study documents beyond doubt that m6dA in human DNA has biological effects”, they authors cannot show the presence of the orthologs of bacterial m6dA methyltransferases (EcoDam and CcrM) in mammals. Therefore, they cannot prove these observations are physiologically relevant. This is just a possibility. They should mitigate their claim.

Reply: Thank you for this hint. We have changed writing at several places now indicating that the methylation was introduced artificially and we only identify potential functions m6dA could have in human cells.

3. Figure 3 shows CCrM-caused difference in gene expression. This difference is simply attributed to the Genome-wide deposition of m6dA by the bacterial CcrM MTase. However, they did not show the direct detection of m6dA in these DE genes. They should show experimental evidence on the presence of m6dA in these genes, especially the position.

Reply: All DE genes contain GANTC sites in their promoters. This point has now been mentioned in the manuscript and we also report the distribution of sites (average 7.1 ± 3.2 SD). The restriction digestion analyses (Suppl. Fig. 1 and 2) document a full genome-wide methylation of GANTC and GATC motifs after the expression of the corresponding MTases. This finding was further supported by the mass spectrometry data (Fig. 2B). This point has been clarified in the manuscript. This result implies that the GANTC motifs associated with DE genes are methylated. This conclusion has now been further validated in newly added restriction protection-qPCR data showing full methylation of genomic DNA after CcrM expression at the identified m6dA sensitive promoters (EMILIN2, ECM1, FAM46B, EGF). These data are now presented in the new Suppl. Fig. 6. Given the genome-wide deposition of m6dA in the target motifs, further m6dA mapping would be pointless as it would only reproduce the occurrence patterns of GANTC motifs.

“4. It is puzzling that, although extremely high m6dA deposited on human DNA, the regulated genes are very few (99 genes). In bacteria, m6dA mainly guide gene replication and DNA mismatch repair. Is it possible that the deposited m6dA may affect DNA replication and repair rather than regulate gene expression?”

Reply: Thank you. We had similar thoughts, but the profile of the mis-regulated genes gives no hints towards DNA replication or DNA repair. As we believe problems in one of these fundamental processes would lead to characteristic secondary responses in gene expression patterns, we cannot propose that these processes are altered on the basis of our data.

“Minor

1. “this modification has been rediscovered in 2015 at low levels in several higher eukaryotes 4.”
Original Cell papers should be cited rather than the cited review article.”

Reply: These references are now given as well. Due to the very controversial situation, we believe providing some reviews that summarize the state of the field is of high relevance for readers as well.

“2. One latest paper regarding 6mdA misincorporation is missed (Lyu et al., Cell Discovery, 8:39 (2022)). This paper shows that misincorporated m6dA is a potential hallmark for human glioblastoma.”

Reply: Thank you. This work has now been cited as well. We also added two more citations, which escaped our attention in the first round of writing.

Li, Z., et al., Nature, 2020. 583(7817): p. 625-630.

Fernandes, S.B., et al., Front Genet, 2021. 12: p. 657171.

REVIEWERS' COMMENTS:

Reviewer #1 (Remarks to the Author):

I appreciate the authors' efforts to answer my concerns, however this referee is still not convinced about the relevance of this work. As stated and acknowledge by the authors, this is a very artificial system where 6-mA is deposited at high frequency on a genome where endogenous 6-mA was not found, therefore not surprisingly affecting gene expression and cell biology. Although it seems evident that the effects observed are attributed to 6-mA, this does not unravel "molecular mechanisms of action endogenous m6dA can have in human cells". If, by extension, one were to overexpress MSssI, a bacterial CpG Methyltransferase, in drosophila cells lacking cytosine methylation, these cells would probably also display altered biology and gene expression, regardless if this modification exists or does not exist.

I would be cautious to publish this manuscript in Communications biology unless the authors mitigate their conclusions by clearly stating that these results do not prove that 6-mA is endogenously present and functional in H293 cells.

Reviewer #2 (Remarks to the Author):

The authors have adequately addressed my previous comments and the manuscript is significantly improved.

Reviewer #3 (Remarks to the Author):

I do not feel the revisions have addressed the lack of biological relevance of this study nor improves our understanding of the physiological relevance (if any) of 6mA in humans. (but this may reflect my biases).

That said, the authors have made a substantial and honest effort to address my concerns including toning down the manuscript, clarifications of methodological issues and highlighting the weaknesses of the study.

As such, I feel the manuscript can be published.

However, I suggest that the raw data is made available in GEO or ArrayExpress. Why is the data being held in a University of Stuttgart server? Once again, the link did not allow access to the data, but directed me to another website (DaRUS-ViPLab-Connector) which gave an error.

Reviewer #4 (Remarks to the Author):

The authors addressed all my questions. I don't have additional comments.

Response to the reviewers' comments COMMSBIO-22-2666B-Z

Thank you for working with our manuscript. There were no open questions left.